# EPiC: Towards Lossless Speedup for Reasoning Training through Edge-Preserving CoT Condensation

## Abstract

Large language models (LLMs) have shown remarkable reasoning capabilities when trained with chain-of-thought (CoT) supervision. However, the long and verbose CoT traces, especially those distilled from large reasoning models (LRMs) such as DeepSeek-R1, significantly increase training costs during the distillation process, where a non-reasoning base model is taught to replicate the reasoning behavior of an LRM. In this work, we study the problem of *CoT condensation* for resource-efficient reasoning training, aimed at pruning intermediate reasoning steps (*i.e.*, thoughts) in CoT traces, enabling supervised model training on length-reduced CoT data while preserving both answer accuracy and the model's ability to generate coherent reasoning. Our rationale is that CoT traces typically follow a three-stage structure: problem understanding, exploration, and solution convergence. Through empirical analysis, we find that retaining the structure of the reasoning trace, especially the early stage of problem understanding (rich in reflective cues) and the final stage of solution convergence (which closely relates to the final answer), is sufficient to achieve *lossless* reasoning supervision. To this end, we propose an Edge-Preserving Condensation method, **EPiC**, which selectively retains only the initial and final segments of each CoT trace while discarding the middle portion. This design draws an analogy to preserving the "edge" of a reasoning trajectory, capturing both the initial problem framing and the final answer synthesis, to maintain logical continuity. Our analyses leveraging the CoT landscape and measuring the mutual information between CoT steps provide further validation for this design. Experiments across multiple model families (Qwen and LLaMA) and benchmarks show that EPiC reduces training time by over 34% while achieving lossless reasoning accuracy (*e.g.*, on Math500), comparable to full CoT supervision. Additionally, we show that EPiC outperforms other condensation methods, including teacher-guided regeneration of condensed CoTs.

## 1 Introduction

LLMs have demonstrated strong performance on complex reasoning tasks, especially when trained with chain-of-thought (CoT) supervision (Wei et al., 2022; Lightman et al., 2023; Guo et al., 2025). CoT training encourages models to generate step-by-step intermediate reasoning before producing a final answer, enhancing both interpretability and task performance in domains such as mathematics (OpenAI et al., 2024) and science (Rein et al., 2024). More recently, large reasoning models (LRMs) such as DeepSeek-R1 (Guo et al., 2025), OpenAI-O1 (OpenAI et al., 2024), and Kimi (Team et al., 2025) have pushed this paradigm further by generating rich CoT traces infused with self-reflection, verification, and backtracking, *e.g.*, acquired via reinforcement learning. These LRMs have enabled a new training pipeline: Their reasoning ability can be distilled into smaller LLMs via supervised fine-tuning (SFT) on LRM-generated CoT data (Guo et al., 2025; Face, 2025; Team, 2025b; Muennighoff et al., 2025; Ye et al., 2025). Throughout this paper, we refer to training (non-reasoning) LLMs with CoT supervision (for reasoning enhancement) as ***reasoning training***.

However, despite their quality, LRM-generated CoT traces are often excessively verbose and suffer from *overthinking*, a tendency to include repetitive or speculative reasoning steps that inflate sequence length without improving final answer accuracy (Chen et al., 2024; Wang et al., 2025).

This verbosity leads to two key issues: (1) high computational cost during SFT, and (2) reduced supervision quality due to noise, particularly in the middle of the trace where speculative exploration dominates. These observations raise a central question: *Are all reasoning steps equally important for reasoning training?*

In this work, we propose Edge-Preserving Condensation (EPiC), a simple yet effective thought-level pruning method that retains the head and tail segments of each CoT trace, corresponding to problem understanding and solution convergence, while removing only the middle portion of the reasoning trajectory. As illustrated in **Figure 1 (left)**, EPiC targets the overgenerated middle stage, preserving the structural and semantic integrity of the reasoning process. As shown in **Figure 1 (right)**, EPiC enables models to achieve competitive reasoning accuracy while reducing training time by *1.5×* compared to full-trace fine-tuning.

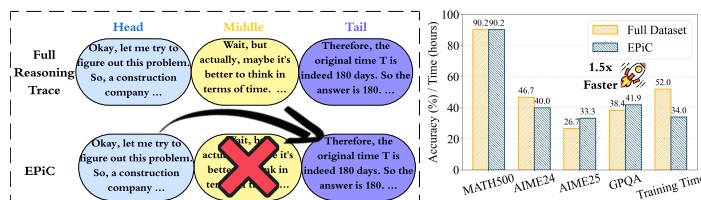

Figure 1: Overview of EPiC. **Left:** EPiC removes the middle portion of CoT while preserving the head (problem understanding) and tail (solution convergence). EPiC applies to training data in OpenR1Math. **Right:** Performance and training time comparison between EPiC and full CoT training based on QWEN2.5-MATH-7B-INSTRUCT. At 50% condensation ratio, EPiC achieves competitive accuracy with 1.5× faster training.

To better understand which segments of a CoT are most critical for reasoning supervision, we conduct a mutual information (MI) analysis between individual segments and the full reasoning trace. We find that the portions selected by EPiC consistently exhibit the highest MI with the complete trace, supporting our empirical finding that the middle segment is the least informative (and often the noisiest) part of the reasoning trajectory. These findings motivate EPiC as a principled and efficient strategy for CoT data-based reasoning training. Our **contributions** are summarized as follows:

• We introduce the first framework EPiC to perform thought-level condensation during training (rather than test-time computing), enabling efficient reasoning distillation by pruning uninformative steps within CoT traces.

• We provide a series of analyses, including thought landscape visualization, mutual information analysis, and CoT perturbation studies, to quantify informativeness across CoT segments and validate the effectiveness of EPiC.

• We conduct extensive experiments across two CoT training datasets (OpenR1Math and GeneralThought), four reasoning benchmarks (MATH500, AIME24/25, and GPQA-Diamond), and multiple non-reasoning model initializations (QWEN2.5-MATH-7B-INSTRUCT, QWEN2.5-7B-INSTRUCT, and LLAMA3.1-8B-INSTRUCT). Our results show that EPiC matches or exceeds the accuracy of full CoT supervision while substantially reducing training time.

## 2 RELATED WORK

**Model distillation for LRMs.** CoT prompting has been shown to significantly improve the reasoning capabilities of large language models (LLMs) (Wei et al., 2022), motivating a line of work that seeks to build LRMs through CoT-style data construction (Zhou et al., 2022; Shridhar et al., 2023; Fu et al., 2023). With the emergence of strong LRMs such as OPENAI-O1 (OpenAI et al., 2024), DEEPSEEK-R1 (Guo et al., 2025), and KIMI-1.5 (Team et al., 2025), which can autonomously generate long and structured CoT traces, including self-reflection, verification, and backtracking, researchers have increasingly focused on distilling such behaviors into smaller models. Guo et al. (2025) was among the first to demonstrate that the reasoning capabilities of LRMs can be effectively transferred to smaller models through SFT. Building on this insight, numerous works have explored distillation from LRM-generated CoT data to improve reasoning performance in smaller LLMs (Team, 2025b; Muennighoff et al., 2025; Ye et al., 2025; Face, 2025; GeneralReasoning, 2024; Team, 2025a; Labs, 2025; Hicham Badri, 2025; Li et al., 2025; Xu et al., 2025; Ji et al., 2025). These works can be broadly grouped into two categories: (1) distillation via high-quality, long-form CoT traces generated from LRMs (Team, 2025b; Muennighoff et al., 2025; Ye et al., 2025; Face, 2025; GeneralReasoning, 2024; Team, 2025a; Labs, 2025; Li et al., 2025; Xu et al., 2025); and (2)

alignment-based approaches that directly supervise logits (Hicham Badri, 2025). A complementary approach, proposed by Ji et al. (2025), combines truncated CoT prefixes with a subset of full traces for more efficient distillation. While prior work has successfully leveraged LRM-generated traces for performance improvement, only a few efforts (Muennighoff et al., 2025; Ye et al., 2025) have addressed the efficiency bottlenecks in CoT distillation. While prior work reduces the number of training examples, our approach retains all examples but shortens each trace through structured thought-level pruning, reducing training cost without compromising model performance.

**Scaling test-time reasoning and the challenge of overlength generation.** Increasing test-time computation has consistently improved model performance on complex reasoning tasks such as mathematical problem solving and code generation (Wei et al., 2023; Wu et al., 2024; DeepSeek-AI et al., 2025; Snell et al., 2024). These gains often come from generating longer reasoning traces or sampling diverse reasoning paths (OpenAI et al., 2024; Wu et al., 2024). Recent methods include parallel path sampling (Wang et al., 2023; Aggarwal et al., 2023; Brown et al., 2024), tree-based search (Yao et al., 2023; Xin et al., 2024), and iterative refinement (Welleck et al., 2023; Madaan et al., 2023; Welleck et al., 2024). Additionally, Muennighoff et al. (2025) proposed enhancing the use of reflection tokens at inference time, and others (Snell et al., 2024; Liu et al., 2025) showed that scaling test-time computation can rival or exceed model size increases. However, these strategies often induce overthinking, verbose, repetitive outputs that slow inference and may reduce quality (Chen et al., 2024; Wang et al., 2025). This is especially common in LRMs, which tend to generate redundant reasoning steps and excessive self-reflection. To mitigate this, several methods promote concise, efficient reasoning: Team et al. (2025), Aggarwal & Welleck (2025), and Luo et al. (2025) introduced length-regularized RL; Xia et al. (2025) apply SFT with truncated or token-pruned inputs; Wang et al. (2025) penalize reflection token usage; and Zhang et al. (2025) compress thoughts via token projection for faster decoding. While prior work mainly targets inference-time efficiency, we focus on training-time efficiency by condensing reasoning trajectories during supervised fine-tuning, enabling smaller models to acquire LRM-style reasoning at lower cost.

**Dataset pruning for efficient training.** To reduce training costs, data pruning has been widely studied in discriminative settings like image classification, where redundant samples are removed (Kothawade et al., 2021; Killamsetty et al., 2021; Lee et al., 2021; Azeemi et al., 2022). Importance score for each sample is estimated using geometry-based (Agarwal et al., 2020), uncertainty-based (Coleman et al., 2019), margin-based (Park et al., 2022), gradient-based (Mirzasoleiman et al., 2020), forgetting-based (Toneva et al., 2018), and training-dynamics-based methods (Paul et al., 2021), with learned pruners also explored (Huang et al., 2023). These approaches have recently been adapted for LLM instruction tuning (Zhang et al.; Xia et al., 2024), and Zhou et al. (2023) showed that strong performance can be achieved with just 1,000 high-quality examples. Pruning for reasoning training remains underexplored. While Ye et al. (2025) used a small curated set of CoT traces, no prior work has examined pruning at the level of individual reasoning steps. In contrast, we propose *thought-level condensation*, a fine-grained strategy that prunes within examples rather than across them.

## 3 Condensed CoT for Efficient Reasoning Training: Motivation and Problem

In this section, we first reviews CoT-based reasoning training and its standard setup, then highlight the trade-off between efficiency and accuracy identified in prior work. Motivated by this tension, we investigate the potential of *thought selection* and formally introduce the problem of *CoT condensation*, which seeks to accelerate reasoning training without compromising reasoning performance.

**Reasoning enhancement via supervised fine-tuning (SFT) on CoT data.** Training LLMs to reason step by step, rather than directly predicting final answers, using CoT supervision has shown significant promise (Guo et al., 2025; Jaech et al., 2024; Xu et al., 2025; Min et al., 2024). Reasoning-based training has proven effective for distillation, allowing smaller non-reasoning LLMs to acquire reasoning skills by fine-tuning on long CoT traces from larger teachers or existing reasoning datasets. In this work, we assume access only to the training CoT dataset, without relying on an additional teacher model for data generation.

Our goal is to improve the efficiency of reasoning training with CoT supervision, achieving faster training while maintaining or even enhancing reasoning capabilities, as measured by final answer

accuracy on complex problems (*e.g.*, mathematics) and the ability to generate coherent reasoning traces (*e.g.*, reflected in the length of reasoning outputs).

To be more concrete, let $\mathcal{D} = \{(\mathbf{x}, \mathbf{r}, \mathbf{y})\}$ denote a CoT-style training dataset, where $\mathbf{x}$ is the input question, $\mathbf{r} = [r_1, r_2, \ldots, r_n]$ denotes the corresponding full reasoning trajectory consisting of $n$ intermediate steps (*i.e.*, thoughts), and $\mathbf{y}$ is the final answer. Following Zhang et al. (2025), we use "\n\n" as a delimiter to simply segment the CoT trajectory $\mathbf{r}$ into different thoughts $\{r_i\}$. In addition, let $\boldsymbol{\theta}$ denote the parameters of an LLM, and let $\pi_{\boldsymbol{\theta}}(\mathbf{b} \mid \mathbf{a})$ represent the model's predicted probability of generating response $\mathbf{b}$ given input $\mathbf{a}$. The reasoning training for $\boldsymbol{\theta}$ under $\mathcal{D}$ becomes

$$\underset{\boldsymbol{\theta}}{\text{minimize}} \quad -\mathbb{E}_{(\mathbf{x},\mathbf{r},\mathbf{y})\in\mathcal{D}} \left[\log \pi_{\boldsymbol{\theta}}(\mathbf{r}, \mathbf{y} \mid \mathbf{x})\right], \tag{1}$$

where the training objective is defined as a cross-entropy sequence prediction loss, which maximizes the likelihood of generating the reasoning trace and final answer conditioned on the input.

**Prior work: Efficiency-accuracy trade-off through dataset size reduction.** While SFT on long CoT significantly enhances the reasoning abilities of LLMs, it is highly resource-intensive, particularly when the traces are generated by LRMs like DEEPSEEK-R1. This renders solving problem (1) computationally expensive, particularly in resource-constrained settings such as academic labs.

To improve the efficiency of reasoning training, *prior work has explored size-reduced, high-quality CoT datasets* such as S1 (Muennighoff et al., 2025) and LIMO (Ye et al., 2025), each containing around 1k carefully curated examples. However, we find that these datasets are typically benchmarked on large models (*e.g.*, 32B), and their effectiveness does not consistently transfer to the training of smaller models. As shown in **Figure 2**, training a 7B model on LIMO or S1 significantly speeds up reasoning training compared to conventional SFT using the larger CoT dataset OpenR1Math (93k examples). However, this speedup comes at the cost of reduced accuracy: 80.0% and 83.6% on MATH500 when using LIMO and S1, respectively, compared to 90.2% when training on OpenR1Math. This indicates that small-scale datasets like S1 and LIMO are insufficient to consistently support effective reasoning performance.

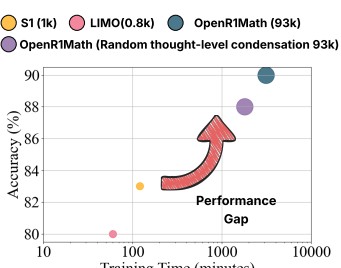

Figure 2: Accuracy and training time for reasoning training on OpenR1Math (93k examples), LIMO (0.8k examples), and S1 (1k examples), using QWEN2.5-MATH-7B-INSTRUCT as the base non-reasoning LLM. Accuracy is evaluated on the MATH500 benchmark. In addition to standard CoT datasets, we also include a thought-level condensed version of OpenR1Math, where 50% of the intermediate thoughts in each CoT trace are randomly retained and the remainder pruned.

**Problem statement.** As motivated by Figure 2, curating smaller CoT datasets does not appear to be an effective solution for improving the efficiency of reasoning training while preserving reasoning performance. Therefore, we propose shifting the focus from *reducing the number of training examples* to *condensing the reasoning trajectory within each example*. We ask whether *thought-level* condensation, rather than example-level reduction, enables more efficient and effective reasoning training.

Therefore, we define the CoT condensation operation as the selection (or pruning) of intermediate thoughts within a reasoning trajectory. Given a CoT trace $\mathbf{r} = [r_1, r_2, \ldots, r_n]$, the condensed version is denoted as $\mathbf{r}_{\text{cond}} = [r_i]_{i\in\Omega}$, where $\Omega \subseteq \{1, \ldots, n\}$ is the index set of selected thoughts and the remaining thoughts are discarded. The potential of thought-level CoT condensation is evident with random thought selection. As shown in Figure 2, randomly selecting 50% of the CoT steps in each example from OpenR1Math has been able to yield 88.0% accuracy, outperforming LIMO and S1, while reducing training time by approximately 40% compared to training on full OpenR1Math.

Figure 2 motivates the central research question of our work: *Can we design an effective CoT condensation method to address the supervised fine-tuning problem in (1), one that substantially reduces training cost while preserving reasoning performance comparable to full-length CoT supervision?*

## 4 EDGE-PRESERVING COT CONDENSATION: METHOD AND RATIONALE

In this section, we begin with a warm-up study to motivate why individual thoughts (*i.e.*, reasoning steps) serve as a proper unit for condensing CoT traces. We then visualize reasoning trajectories generated by a LRM and observe that the middle portions of these trajectories often drift away from

the correct final answer. This insight motivates our proposed method, EPiC, which performs CoT condensation by explicitly leveraging the structural characteristics of reasoning trajectories. Finally, we justify the design of EPiC from two complementary perspectives: (1) the mutual information between individual reasoning steps and the final answer, and (2) a sensitivity analysis that contrasts the importance of reasoning structure versus content.

**CoT condensation unit: thoughts or tokens?** As an alternative to thought-level condensation, one may consider pruning a CoT trace at the token level. A representative approach is **TokenSkip** (Xia et al., 2025) to assign importance scores to individual tokens and prune those deemed less critical (Pan et al., 2024). We can apply this method to compress CoT traces. However, when training models on these token-pruned CoT datasets, we observe a significant drop in performance compared to training on the original, unpruned data, as shown in **Figure 3**. Peering into the results, we find that token-level pruning disrupts thought-level reasoning patterns, producing fragmented and grammatically broken inputs. As validated in Fig. 3(Left), TokenSkip removes transitional markers and reflective words (*e.g.*, "wait"), which are crucial

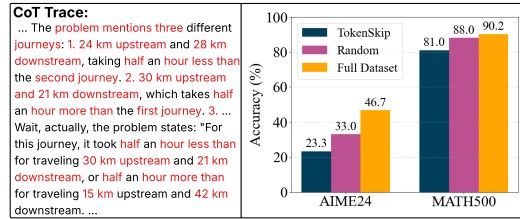

Figure 3: Performance of TokenSkip-based token-level condensation for reasoning training. *(Left)* Visualization of a CoT trace pruned by TokenSkip (Xia et al., 2025) with a 50% pruning ratio. Tokens highlighted in red are retained, while the rest are removed. *(Right)* Final answer accuracy of models trained on three datasets: TokenSkip-pruned (50%), random thought-level condensation (50%), and the original full dataset, evaluated on AIME24 and MATH500. All models are fine-tuned from QWEN2.5-MATH-7B-INSTRUCT on the OpenR1Math dataset.

for connecting thoughts and preserving logical flow. The resulting sentences are syntactically flawed and risk confusing the language model during training. Therefore, token-level (*e.g.*, CoT step-wise) condensation becomes *in*effective for reasoning training, performing even worse than random thought-level condensation in Figure 3(Right).

**Drift of middle reasoning steps in LRMs.** To identify which parts of a reasoning trajectory are less effective for training, we analyze reasoning traces using the CoT landscape visualization tool (Zhou et al., 2025). This tool projects trajectories into a latent semantic space, providing an interpretable view of how individual steps relate to the correct answer. As shown in **Figure 4**, darker regions denote states semantically *closer* to the correct answer, with the $x$ and $y$ axes representing two t-SNE–projected dimensions. The visualization reveals that many intermediate steps drift away from the correct answer, even when initial steps start closer to the solution path. Although the final steps eventually converge, these intermediate steps introduce "noise", which misguide the model away from the correct trajectory. These observations suggest that the middle portion of a CoT trace is often less informative, or even detrimental, to reasoning accuracy, motivating a stage-wise perspective that examines the distinct contributions of different trajectory segments.

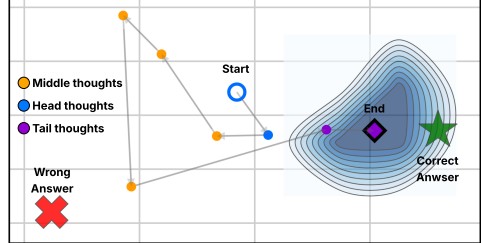

Figure 4: Visualization of a reasoning trajectory generated by DEEPSEEK-R1-DISTILL-QWEN-7B on the AQuA (Ling et al., 2017) dataset. The plot is produced using the trajectory landscape tool from (Zhou et al., 2025), where each node represents the model's reasoning state in a latent space after $k$ thought steps. And $x$ and $y$ axes correspond to two t-SNE-projected dimensions. The trajectory is segmented into three parts: the first 25% of steps (blue, "head thoughts"), the middle 50% (orange, "middle thoughts"), and the final 25% (violet, "tail thoughts"). The correct answer is shown as a green star, while red cross denotes incorrect (distractor) answers.

**Edge-preserving condensation (EPiC).** Building on the earlier insight that intermediate reasoning steps may contribute little to final answer, and that per-sample graph analysis (Zhou et al., 2025) is too costly for large datasets, we propose a structure-based condensation method. We roughly view each CoT trace as consisting of three stages: the beginning, middle, and end of the trajectory. These stages generally serve different functional roles in reasoning. The beginning stage, *Understand*, involves parsing and interpreting the problem; the middle stage, *Explore*, entails inferring and iterating through possible reasoning paths; and the tail stage, *Converge*, synthesizes information and finalizes the solution. An illustration of this three-stage structure is provided in **Figure A1**. Reasoning condensation can be realized by selectively removing one of these stages from the full trajectory.

Based on the above CoT segmentation, we next develop EPiC, a method that preserves only the head and tail portions of the CoT trace, effectively connecting the initial and final stages while discarding the exploration stage. This design mirrors the idea of retaining the "edges" of a reasoning trajectory. Recall that $\mathbf{r} = [r_1, r_2, \ldots, r_n]$ denotes the full reasoning trajectory consisting of $n$ thoughts. We define the *condensation ratio* (**CR**) $\tau \in [0,1]$ as the fraction of thoughts retained after pruning (*i.e.*, the length of the condensed trajectory). EPiC compresses the full trajectory $\mathbf{r}$ into $\mathbf{r}_{\text{cond}}$ by pruning the middle portion while retaining the head and tail:

$$\mathbf{r}_{\text{cond}} = [r_i]_{i \in \Omega}, \quad \Omega = \left\{1, \ldots, \left\lfloor \frac{\tau n}{2} \right\rfloor\right\} \cup \left\{n - \left\lfloor \frac{\tau n}{2} \right\rfloor + 1, \ldots, n\right\}. \tag{EPiC}$$

Here, $\lfloor \cdot \rfloor$ denotes the floor function. The total number of retained thoughts, $\lfloor \tau n \rfloor$, is equally divided between the head and tail segments, each of length $\left\lfloor \frac{\tau n}{2} \right\rfloor$. Please refer to Appendix B for visualizations of example reasoning traces after condensation.

**Understanding EPiC via mutual information (MI).** To further understand which parts of the reasoning trajectory are most important for improving reasoning ability, we analyze EPiC using MI. Our goal is to quantify how much information different portions of the reasoning trace retain compared with the full reasoning trace. For a condensed trace $\mathbf{r}_{\text{cond}} = [r_i]_{i \in \Omega}$, we obtain a matrix representation $\mathbf{E}_\Omega = [\mathbf{e}_1^\Omega, \ldots, \mathbf{e}_m^\Omega]^\top \in \mathbb{R}^{m \times d}$ by feeding each trace through a pretrained LLM and applying mean pooling over the final hidden states across the token dimension. Here, $m$ denotes the number of samples used for MI evaluation and $d$ is the hidden dimension of the model. We compute the mutual information between $\mathbf{E}_\Omega$

Table 1: Comparison of MI, computed using (A1), between the full reasoning trajectory and selected portions of the reasoning trajectory under various condensation methods and condensation ratios ($\tau$). The evaluation is performed on 2500 examples sampled from the OpenR1Math dataset using the QWEN2.5-1.5B-INSTRUCT model.

| Method | $\tau = 0.01$ | $\tau = 0.05$ | $\tau = 0.1$ | $\tau = 0.5$ |
|---|---|---|---|---|
| Full ($\tau = 1$) | | 8.77 | | |
| Random | 0.56 | 1.90 | 2.64 | 4.57 |
| HoC | 0.93 | 1.77 | 2.27 | 4.85 |
| MoC | 0.64 | 1.39 | 1.84 | 3.81 |
| ToC | 0.43 | 1.06 | 1.46 | 3.05 |
| EPiC | 3.07 | 3.57 | 4.06 | 8.70 |

and $\mathbf{E}_{\text{Full}}$, denoted as $\mathcal{I}(\mathbf{E}_\Omega; \mathbf{E}_{\text{Full}})$, using the Kraskov estimator Kraskov et al. (2004), which approximates MI based on distances between nearest neighbors in the sample space. See Appendix C for more details. The MI score serves as a proxy for how informative the selected reasoning steps are compared with the full reasoning trace.

A higher $\mathcal{I}(\mathbf{E}_\Omega; \mathbf{E}_{\text{Full}})$ indicates that the condensed subset $\Omega$ preserves more of the information in the full reasoning trace, and thus corresponds to a more effective condensation strategy. We compute MI between the full reasoning trace and different portions of the reasoning trajectory to assess the informativeness of each segment. Specifically, given a CR (condensation ratio) $\tau$, we define: *Head-only Condensation* (**HoC**) as $\Omega_{\text{H}} = \{1, \ldots, \lfloor \tau n \rfloor\}$, *Tail-only Condensation* (**ToC**) as $\Omega_{\text{T}} = \{n - \lfloor \tau n \rfloor + 1, \ldots, n\}$, and *Middle-only Condensation* (**MoC**) as $\Omega_{\text{M}} = \left\{\left\lfloor \frac{(1-\tau)n}{2} \right\rfloor + 1, \ldots, n - \left\lfloor \frac{(1-\tau)n}{2} \right\rfloor\right\}$. As shown in **Table 1**, EPiC consistently achieves the highest MI across all condensation ratios $\tau \in \{0.01, 0.05, 0.1, 0.5\}$, closely matching the MI of the full reasoning trace. This indi-

Figure 5: Final answer accuracy comparison for reasoning training using QWEN2.5-MATH-7B-INSTRUCT on various perturbed CoT training sets, evaluated on AIME24 and MATH500 at test time. Perturbations are applied to specific regions of the CoT trace–head, tail, middle, entire trace-or not applied at all (no perturbation).

cates that EPiC effectively preserves the most informative parts of the reasoning trace. Notably, at $\tau = 0.5$, EPiC attains an MI of 8.70, nearly matching the full trace MI of 8.77. This indicates that EPiC preserves nearly all the semantic content of the full reasoning trajectory while using only half the tokens, providing strong evidence that its structural selection strategy captures the most informative parts of the trace. We observe similar results using QWEN2.5-MATH-7B-INSTRUCT, as shown in Table A2.

**Rationalizing EPiC through reasoning structure vs. content.** To validate the importance of the CoT stages identified by EPiC, we perform a perturbation analysis. Instead of removing the reasoning steps outside the condensation set $\Omega$, we replace their content with randomly sampled text while preserving the overall structural layout of the trace. During this perturbation, we retain reflection tokens, often realized as discourse markers such as `wait` and `hmm` (Muennighoff et al., 2025), since they provide transitional and reflective cues that help maintain coherence between thoughts.

Building on the above, we investigate the impact of fixing the reasoning condensation pattern $\Omega$ while perturbing the unselected thoughts $\{r_i\}_{i \notin \Omega}$ by replacing their content (between reflection

tokens) with randomly sampled sentences from WikiText (Merity et al., 2016). **Figure 5** shows how reasoning training on the perturbed CoT dataset impacts model performance. xWe evaluate five settings: perturbing (1) all reasoning steps, or 50% of the trace from the (2) head, (3) middle, or (4) tail segments. Perturbing the middle yields the smallest degradation, with only a slight drop in accuracy compared to the original dataset. In contrast, perturbing the head or tail causes larger declines, while perturbing the entire trace severely harms performance. These results support the core idea of EPiC: *the middle stage of a CoT trace is less critical than the head and tail, and much of it can be pruned or perturbed without substantially compromising reasoning ability.*

## 5 EXPERIMENTS

### 5.1 EXPERIMENT SETUPS

**Training datasets.** To demonstrate the effectiveness of EPiC in facilitating CoT training for enhanced reasoning capabilities, we train models on two long-form CoT datasets distilled from DEEPSEEK-R1: **(1) OpenR1Math** (Face, 2025): This dataset comprises 220k math problems, each paired with reasoning traces generated by DEEPSEEK-R1. Answers are verified using either a math verifier (Kydlíček, 2024) or LLAMA-3.3-70B-INSTRUCT to ensure correctness. In our experiments, we use the default main subset, which includes 93k verified examples. **(2) GeneralThoughts** (GeneralReasoning, 2024): This dataset offers a diverse reasoning traces beyond mathematics and coding, spanning natural sciences, humanities, social sciences, and general conversational reasoning. The traces are generated by a diverse set of strong LLMs, including O3-MINI, GEMINI-2-FLASH-THINKING, CLAUDE-3.7-SONNET, and DEEPSEEK-R1.

**Model setups.** In our experiments, we primarily use the *non-reasoning* LLM QWEN2.5-MATH-7B-INSTRUCT (Yang et al., 2024b) as the base model for SFT-based reasoning training, due to its strong mathematical capabilities. To evaluate the robustness and generalizability of EPiC across different model initializations, we additionally conduct experiments with two alternative models: QWEN2.5-7B-INSTRUCT (Yang et al., 2024a), which shares the same architecture but lacks math-specific instruction tuning, and LLAMA3.1-8B-INSTRUCT (Grattafiori et al., 2024), which differs in both architecture and pretraining corpus. These variants assess EPiC's effectiveness when initialized from a weaker math model or a different architecture. Given our computing resources, we focus on 7B/8B-scale models. We exclude much smaller models (like 1.5B) due to their limited capability and instability in reasoning training.

**Evaluation benchmarks.** To assess the acquired reasoning capabilities, we primarily evaluate models on three benchmarks: **(1) MATH500** Lightman et al. (2023): A curated set of 500 multi-step problems from the OpenAI MATH benchmark, designed to measure mathematical reasoning ability. **(2) AIME24/25** (MAA Committees): Two separate benchmarks, each containing 30 high school competition-level mathematics problems from the 2024 and 2025 American Invitational Mathematics Examination (AIME), respectively. **(3) GPQA-Diamond** (Rein et al., 2024): A graduate-level STEM benchmark consisting of multiple-choice questions in biology, physics, and chemistry. All problems are written and verified by domain experts (PhD-level), providing a challenging testbed for evaluating general scientific reasoning beyond mathematics. For evaluation, we set a maximum generation length of 9000 tokens for both MATH500 and AIME24, and 4000 tokens for GPQA-DIAMOND. Decoding is performed using nucleus sampling with a temperature of 0.6 and top-$p$ of 0.95, following (Guo et al., 2025). In addition to final answer accuracy, we also assess reasoning generation quality using two auxiliary metrics: (1) the length of the generated reasoning traces, and (2) the number of reflection tokens, which serve as strong indicators of reasoning ability, *e.g.*, the "Aha Moment" emphasized in DEEPSEEK-R1 (Guo et al., 2025).

**Baselines.** To evaluate the effectiveness of EPiC, we compare it against several baseline condensation strategies: **(1) Random (TL):** Randomly selects a subset of reasoning steps per sample at the thought level (TL) according to the condensation ratio $\tau$, ignoring their positions in the trajectory . **(2) Random (DL):** Randomly selects a subset of training samples at the data level (DL) with ratio $\tau$, ensuring comparable training time to other baselines. **(3) HoC (Head-only Condensation):** Retains only the first $\lfloor \tau n \rfloor$ steps of the reasoning trajectory, where $n$ is the total number of steps. **(4) ToC (Tail-only Condensation):** Retains only the last $\lfloor \tau n \rfloor$ steps. **(5) TokenSkip** (Xia et al., 2025): A recent token-level condensation that scores and selects important tokens across the trace for reten-

Table 2: Performance comparison of EPiC against full dataset training and baseline condensation methods across four reasoning benchmarks: Math500, AIME24, AIME25 and GPQA-Diamond. Each benchmark reports both accuracy (%) , the average number of generated tokens (#Toks) and the average number of reflection tokens (#Rtoks). All models are trained via SFT from Qwen2.5-Math-7B-Instruct, using a fixed condensation ratio of 50%. The reasoning training is conducted on two datasets, OPENR1MATH and GENERALTHOUGHT195K, respectively. The final column reports the total training time in hours.

| Methods | Math500 | | | AIME24 | | | AIME25 | | | GPQA-Diamond | | | Time |
|---|---|---|---|---|---|---|---|---|---|---|---|---|---|
| | Acc | #Toks | #RToks | Acc | #Toks | #RToks | Acc | #Toks | #RToks | Acc | #Toks | #RToks | (Hours) |
| w/o SFT | 82.6 | 696.6 | 0.0 | 3.3 | 1624.2 | 0.1 | 6.67 | 1444.5 | 0.0 | 34.9 | 1331.5 | 0.0 | - |
| **OpenR1Math** (Face, 2025) | | | | | | | | | | | | | |
| Full dataset | 90.2 | 3213.5 | 17.6 | 46.7 | 7365.2 | 43.7 | 26.7 | 7544.4 | 49.0 | 38.4 | 3817.1 | 37.0 | 51.9 |
| Tokenskip | 81.0 | 4861.5 | 28.0 | 23.3 | 8499.4 | 48.3 | 6.7 | 6559.0 | 0.0 | 31.3 | 3896.7 | 30.4 | 30.0 |
| Random (TL) | 88.0 | 3221.9 | 17.2 | 33.3 | 7382.8 | 45.8 | 26.7 | 7417.8 | 53.4 | 36.4 | 3802.7 | 37.8 | 32.4 |
| Random (DL) | 88.6 | 3278.4 | 18.0 | 36.7 | 7880.0 | 44.3 | 23.3 | 7855.0 | 54.7 | 39.3 | 3794.5 | 32.4 | 33.8 |
| HoC | 89.6 | 3178.3 | 17.3 | 33.3 | 7549.0 | 45.9 | 26.7 | 7123.9 | 45.0 | 40.4 | 3753.5 | 39.5 | 34.2 |
| ToC | 84.6 | 3088.6 | 16.4 | 33.3 | 7141.4 | 41.8 | 26.7 | 7384.9 | 46.4 | 43.9 | 3472.8 | 28.7 | 32.0 |
| EPiC | 90.2 | 3109.1 | 17.5 | 40.0 | 7330.8 | 45.4 | 33.3 | 7625.7 | 47.6 | 41.9 | 3725.7 | 37.8 | 34.0 |
| **GeneralThought195k** (GeneralReasoning, 2024) | | | | | | | | | | | | | |
| Full dataset | 87.0 | 3072.7 | 23.3 | 26.7 | 7613.6 | 50.2 | 26.7 | 8017.1 | 52.0 | 40.4 | 3494.9 | 46.2 | 48.8 |
| Tokenskip | 58.4 | 4281.8 | 0.0 | 0.0 | 9000.0 | 0.0 | 6.7 | 8376.7 | 0.0 | 29.3 | 3653.5 | 0.0 | 32.0 |
| Random (TL) | 58.2 | 3621.8 | 27.0 | 0.0 | 7591.9 | 52.0 | 0.0 | 7213.5 | 61.2 | 35.0 | 3329.4 | 41.4 | 32.0 |
| Random (DL) | 85.0 | 2791.8 | 16.2 | 20.0 | 7861.8 | 60.0 | 16.7 | 8208.3 | 66.9 | 41.9 | 3395.9 | 38.3 | 32.9 |
| HoC | 85.8 | 3252.7 | 18.3 | 26.7 | 8004.9 | 43.8 | 23.3 | 7717.8 | 56.2 | 37.4 | 3490.1 | 40.1 | 33.5 |
| ToC | 75.4 | 2963.8 | 19.1 | 13.3 | 6991.3 | 46.6 | 20.0 | 7635.4 | 39.53 | 41.4 | 3182.5 | 31.5 | 31.6 |
| EPiC | 86.0 | 2874.2 | 18.5 | 20.0 | 7967.2 | 46.9 | 26.7 | 7422.3 | 55.2 | 42.4 | 3388.3 | 40.8 | 32.3 |

tion. Unless otherwise specified, the condensation ratio $\tau$ is set to 50% throughout our experiments. Please refer to the training and implementation details in Appendix D.

## 5.2 EXPERIMENT RESULTS

**Performance overview of EPiC vs. full-data training and condensation baselines.** In **Table,2**, we evaluate the performance of EPiC under a 50% condensation ratio across three reasoning benchmarks: MATH500, AIME24/25, and GPQA-DIAMOND. We compare against baselines trained on two datasets: OPENR1MATH, which focuses exclusively on mathematical problems, and GENERALTHOUGHT195K, which contains reasoning traces spanning diverse domains such as science, humanities, and general knowledge. All models are fine-tuned from QWEN2.5-MATH-7B-INSTRUCT.

First, EPiC matches the performance of full-data training while significantly reducing training time up to *34%*. For example, when trained on OPENR1MATH, EPiC achieves 90.2% accuracy on MATH500, identical to the full model, but requires only 34.0 hours of training compared to 51.9 hours for the full dataset. On GENERALTHOUGHT195K, EPiC also maintains comparable performance (86.0% vs. 87.0%) while reducing training time from 48.8 to 32.3 hours, demonstrating substantial efficiency gains without performance loss.

In-domain evaluation on GPQA-DIAMOND further validates the strength of EPiC. Since GENERALTHOUGHT195K includes STEM-related reasoning (*e.g.*, physics and biology), GPQA-DIAMOND serves as a natural in-domain test. As shown, EPiC achieves 42.4% accuracy, outperforming the full-data baseline of 40.4%. Even when treated as an out-of-domain task–training only on math-focused OPENR1MATH–EPiC generalizes better than full-data training (41.9% vs. 38.4%), suggesting that pruning the middle portion of reasoning traces may help improve generalization.

Compared to other structural condensation baselines, EPiC also exhibits clear advantages. Both HoC and ToC, which preserve only the head or tail of the reasoning trace, perform noticeably worse than EPiC across all tasks and datasets. These results confirm that preserving both the beginning and end of the reasoning trace, while discarding the middle, is a highly effective and efficient strategy.

Last but not least, we analyze reasoning behavior through the lens of generation length. Across all benchmarks, EPiC produces responses of *comparable length* to those generated by models trained on the full dataset, indicating that it effectively preserves reasoning complexity. For example, on MATH500, EPiC achieves the same 90.2% accuracy as the full model while generating, on average, only 100 fewer tokens (3109.1 vs. 3213.5). In addition, we observe that EPiC maintains a similar number of reflection tokens, such as "wait", "hmm", and other metacognitive markers, compared to the full model, further confirming that its condensed traces still elicit rich and deliberate reasoning behavior during inference. This result is particularly noteworthy given that a 50% condensation ratio was applied to the CoT training data. Despite this reduction, the model's ability to generate complete and coherent reasoning traces at test time remains essentially *lossless* compared to full-data training.

Appendix F also includes qualitative generation examples illustrate that EPiC achieves comparable reasoning quality to the full-data model.

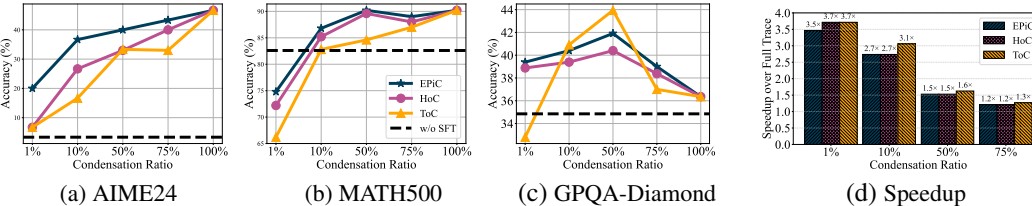

| (a) AIME24 | (b) MATH500 | (c) GPQA-Diamond | (d) Speedup |

Figure 6: Reasoning accuracy of CoT training at different condensation ratios using EPiC, HoC, and ToC, on three benchmarks: **(a)** AIME24, **(b)** MATH500, and **(c)** GPQA-Diamond. All models are fine-tuned using Qwen2.5-Math-7B-Instruct on OpenR1Math. The dashed line indicates performance without SFT, and the 100% condensation ratio refers to the full training dataset. **(d)** shows the training speedup relative to full-trace fine-tuning across condensation ratios for each method.

**Performance against CoT condensation ratios.** In **Figure 6-(a,b,c)**, we present the performance of reasoning training under varying CoT condensation ratios using different condensation methods (EPiC, HoC, ToC), evaluated on the reasoning benchmarks AIME24, MATH500, and GPQA-Diamond. As expected, reasoning performance generally improves as the condensation ratio increases (*i.e.*, more thoughts are retained), where 100% corresponds to the full-data training scenario. Notably, GPQA-Diamond exhibits a sweet spot at 50% condensation, where the accuracy even surpasses that of the full-dataset baseline. This suggests that moderate pruning may help eliminate noisy or redundant reasoning, thereby improving generalization. We also observe that the use of CoT training data plays a key role in driving reasoning accuracy, compared to the initial model without SFT. Across all benchmarks and condensation levels, EPiC consistently outperforms both HoC and ToC, reinforcing the results shown in Table 2. Furthermore, **Figure 6-(d)** reports the corresponding training speedups relative to full CoT fine-tuning. As expected, lower condensation ratios lead to faster training. At 50% condensation, EPiC delivers a substantial $1.5\times$ improvement in training efficiency over full-data training.

**Comparison between EPiC and teacher-guided CoT condensation.** In what follows, we assume access to a teacher LRM capable of generating CoT data and examine whether EPiC continues to outperform when CoT traces are re-generated with controlled lengths. To

Table 3: Performance comparison between EPiC and teacher-guided CoT condensation on the LIMO dataset. Results are reported in the same format as Table 2.

| Methods | Math500 | | AIME24 | | AIME25 | | GPQA Diamond | | Time |
|---|---|---|---|---|---|---|---|---|---|
| | Acc | #Toks | Acc | #Toks | Acc | #Toks | Acc | #Toks | (Hours) |
| Teacher-guided | 78.4 | 2263.3 | 13.3 | 6969.3 | 10.0 | 6876.0 | 34.8 | 3460.8 | 0.7 |
| EPiC | 81.4 | 2680.7 | 16.7 | 7173.9 | 13.3 | 7271.1 | 29.8 | 3748.7 | 0.8 |

this end, we use DEEPSEEK-R1-DISTILL-QWEN-32B as the teacher model to produce reasoning traces for LIMO questions (Ye et al., 2025). Following the S1 approach (Muennighoff et al., 2025), we append "final answer" prompts to elicit complete responses before the token limit, serving as an inference-time baseline for data shortening. For fairness, we match the generated token count to EPiC's 50% condensation ratio relative to the full LIMO dataset and then conduct SFT on QWEN2.5-MATH-7B-INSTRUCT for comparison. **Table 3** shows that EPiC surpasses the teacher-guided baseline (*i.e.*, training the student model on the teacher-distilled, shortened LIMO dataset).

**Additional results.** Beyond the main results, Figure A2 shows that EPiC does not degrade performance on harder problems. Table A3 further demonstrates its robustness across different model initializations, achieving comparable or better accuracy with lower training cost.

## 6  CONCLUSION

In this work, we propose **EPiC**, a simple thought-level condensation strategy that preserves only the head and tail of long CoT traces. Motivated by the redundancy of intermediate steps, we analyze segment informativeness and show that removing the middle portion yields substantial efficiency gains without harming performance. Experiments across benchmarks demonstrate that EPiC matches or exceeds full-data training while reducing cost, and remains robust across datasets, difficulty levels, and model architectures. This provides a practical and interpretable approach to efficient reasoning supervision, an increasingly important need as full CoT training grows more costly. Limitations and broader impacts are discussed in **Appendix G**, **Appendix H**, and details of LLM usage are provided in **Appendix I**.

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

APPENDIX

# A ILLUSTRATION OF THREE-STAGE STRUCTURE OF LONG CoT REASONING

To provide a general illustration of reasoning traces generated by LRMs, we rougly treat each trace into three coarse-grained stages, as shown in Figure A1. This partition reflects the natural progression of problem solving and helps us better interpret the internal dynamics of chain-of-thought reasoning. Specifically, the beginning stage, *Understand*, corresponds to parsing the problem statement, recalling relevant knowledge, and setting up the task to be solved. The middle stage, *Explore*, involves inferring and iterating through possible reasoning paths, often including trial-and-error steps, exploratory calculations. Finally, the tail stage, *Converge*, synthesizes the accumulated information, prunes away unsuccessful paths, and finalizes the reasoning into a coherent solution.

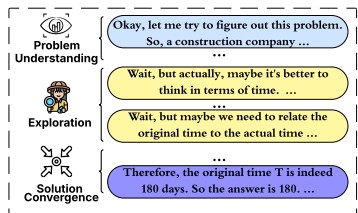

Figure A1: Illustration of three-stage structure of long CoT reasoning: problem understanding (head), exploration (middle), and solution convergence (tail).

# B VISUALIZATION OF CONDENSED REASONING EXAMPLES

To provide qualitative insight into how EPiC condense long reasoning traces, Table A1 presents an example from the condensed training dataset based on OPENR1MATH. This example highlights the head and tail portions retained by EPiC, with the pruned middle segment shown in red.

# C IMPLEMENTATION OF MUTUAL INFORMATION AND EXTENDED ANALYSIS

**Implementation details of mutual information.** To identify which parts of the reasoning trajectory contribute most to model learning, we analyze the mutual information (MI) between different portions of the trace and the full CoT trajectory. For a condensed trace $\mathbf{r}_{\text{cond}} = [r_i]_{i \in \Omega}$, we encode it using a pretrained LLM and apply mean pooling over the token dimension to obtain a representation matrix $\mathbf{E}_\Omega \in \mathbb{R}^{m \times d}$, where $m$ is the number of samples and $d$ is the hidden dimension. We then compute MI between $\mathbf{E}_\Omega$ and the full trace embedding $\mathbf{E}_{\text{Full}}$ as $\mathcal{I}(\mathbf{E}_\Omega; \mathbf{E}_{\text{Full}})$ using the Kraskov $k$-nearest neighbor estimator Kraskov et al. (2004), which approximates MI based on distances between nearest neighbors in the sample space. This non-parametric method is well-suited for high-dimensional representations and provides a robust estimate of MI without requiring density assumptions. We use $k = 5$ in all experiments. For each $i \in \{1, \ldots, m\}$, we compute the radius $\rho_i$ using the $\ell_\infty$ norm as

$$\rho_i = \min_{j \in \mathcal{E}_{i,k}} \max \left\{ \|\mathbf{e}_i^\Omega - \mathbf{e}_j^\Omega\|_\infty, \|\mathbf{e}_i^{\text{Full}} - \mathbf{e}_j^{\text{Full}}\|_\infty \right\},$$

where $\mathbf{e}_i^\Omega$ and $\mathbf{e}_i^{\text{Full}}$ are the $i$th rows of $\mathbf{E}_\Omega$ and $\mathbf{E}_{\text{Full}}$, respectively. $\mathcal{E}_{i,k} \subseteq \{1, \ldots, m\} \setminus \{i\}$ denotes the indices of the $k$-nearest neighbors of the joint embedding $(\mathbf{e}_i^{\text{Full}}, \mathbf{e}_i^\Omega)$ in the joint space $\mathbb{R}^{2d}$. Using this radius $\rho_i$, we then count the number of neighbors of $\mathbf{e}_i^\Omega$ and $\mathbf{e}_i^{\text{Full}}$ that lie within $\rho_i$ in their respective marginal spaces

$$n_i^\Omega = \left| \left\{ j \neq i \,;\, \|\mathbf{e}_i^\Omega - \mathbf{e}_j^\Omega\|_\infty < \rho_i \right\} \right|, \quad n_i^{\text{Full}} = \left| \left\{ j \neq i \,;\, \|\mathbf{e}_i^{\text{Full}} - \mathbf{e}_j^{\text{Full}}\|_\infty < \rho_i \right\} \right|.$$

Finally, we estimate the mutual information as

$$\mathcal{I}(\mathbf{E}_\Omega; \mathbf{E}_{\text{Full}}) = \psi(k) + \psi(m) - \frac{1}{m} \sum_{i=1}^m \left[ \psi(n_i^\Omega + 1) + \psi(n_i^{\text{Full}} + 1) \right], \tag{A1}$$

where $\psi(\cdot)$ is the digamma function. The MI score (A1) serves as a proxy for how informative the selected reasoning steps are compared with the full reasoning trace. **Additional results for mutual information.** To further validate the robustness of our mutual information analysis, we perform an additional evaluation using the model, QWEN2.5-7B-INSTRUCT, to compute the latent representations $\mathbf{E}_\Omega$. As shown in Table A2, EPiC consistently achieves the highest MI across all tested

Table A2: Comparison of MI, computed using (A1), between the full reasoning trajectory and selected portions of the reasoning trajectory under various condensation methods and condensation ratios ($\tau$). The evaluation is performed on 2500 examples sampled from the OpenR1Math dataset using the QWEN2.5-7B-INSTRUCT model.

| Method | $\tau = 0.01$ | $\tau = 0.05$ | $\tau = 0.1$ | $\tau = 0.5$ |
|---|---|---|---|---|
| Full ($\tau = 1$) | | 8.79 | | |
| Random | 0.41 | 1.79 | 2.48 | 4.27 |
| HoC | 1.03 | 1.97 | 2.39 | 4.90 |
| MoC | 0.41 | 1.25 | 1.71 | 3.65 |
| ToC | 0.46 | 1.19 | 1.50 | 3.07 |
| EPiC | 3.11 | 3.58 | 4.07 | 8.67 |

condensation ratios $\tau \in \{0.01, 0.05, 0.1, 0.5\}$, outperforming all other condensation baselines. Remarkably, at $\tau = 0.5$, EPiC attains an MI of 8.67, which is almost indistinguishable from the MI of the full reasoning trace (8.79). These findings are consistent with the results reported in Table 1, and further corroborate that EPiC preserves the majority of semantic content in the reasoning trace while using only 50% of the tokens. This highlights the effectiveness of our method in maintaining reasoning fidelity under significant token budget constraints.

# D    ADDITIONAL EXPERIMENTAL DETAILS

## D.1    TRAINING SETUP

**Supervised fine-tuning setup.** We adopt a unified training configuration across all base models and data condensation strategies to ensure fair comparison. All models are fine-tuned for 3 epochs using the AdamW optimizer with a learning rate of $5 \times 10^{-5}$, weight decay of 0.0001, and a linear learning rate scheduler with 10% warmup. Training is performed on 8 NVIDIA A6000 GPUs with a global batch size of 16, achieved via a per-device batch size of 1 and gradient accumulation over 2 steps. We use `bfloat16` precision and enable gradient checkpointing for memory efficiency. To improve throughput, long sequences are packed into fixed-length inputs with a maximum context length of 32,768 tokens.

## D.2    INFERENCE SETUP

For evaluation, we set a maximum generation length of 9000 tokens for both MATH500 and AIME24, and 4000 tokens for GPQA-DIAMOND. Decoding is performed using nucleus sampling with a temperature of 0.6 and top-$p$ of 0.95, following Guo et al. (2025). For AIME24, we sample 32 responses per query and report *pass@1*. For all other benchmarks, we report accuracy from a single sampled response.

# E    ADDITIONAL EXPERIMENTS

## E.1    PERFORMANCE AGAINST PROBLEM DIFFICULTY LEVELS.

In **Figure A2**, we present a fine-grained breakdown of model performance across five difficulty levels on the Math500 benchmark. The top plot shows accuracy comparisons, while the bottom plot reports the corresponding average number of generated tokens. All models are trained using QWEN2.5-MATH-7B-INSTRUCT on the OpenR1Math dataset. As expected, accuracy generally decreases and generation length increases as problem difficulty rises. This trend holds consistently across all condensation strategies, reflecting the intrinsic complexity of harder problems. However, EPiC performs *better than the full-data baseline* on the most difficult Level 5 problems. This indicates that CoT condensation via EPiC did *not* disproportionately disadvantage on the harder levels. In addition, as evidenced by the bottom plots, all condensation methods applied to the CoT training datasets do not hinder the reasoning generation capability of the resulting models after training, across all problem difficulty levels. This also echoes the finding in Table 2 that reasoning ability can be effectively acquired using shorter CoT traces without compromising generation quality.

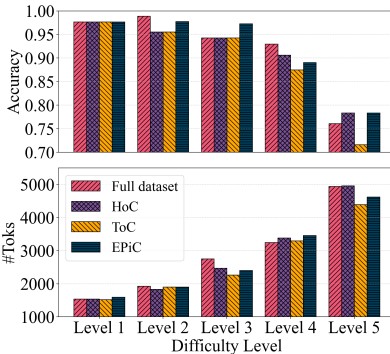

Figure A2: Accuracy and generation length across difficulty levels on the Math500 benchmark. **(Top)** Accuracy comparison of condensation methods (50% condensation ratio) and full-data baseline across five difficulty levels. **(Bottom)** Average number of generated tokens per method and difficulty level. All models are fine-tuned from QWEN2.5-MATH-7B-INSTRUCT on OpenR1Math.

## E.2 EPiC IS RESILIENT TO NON-REASONING BASE MODEL CHOICE.

To evaluate the robustness of EPiC under different initialization conditions, we assess its performance when fine-tuning two distinct pretrained backbones: QWEN2.5-7B-INSTRUCT (Yang et al., 2024a) and LLaMA3.1-8B-INSTRUCT (Grattafiori et al., 2024). As shown in **Table A3**, EPiC consistently achieves strong performance despite using only 50% of the original reasoning traces. Compared to full-data fine-tuning, it attains comparable or even superior accuracy while significantly reducing training time, saving 19.1 hours on LLaMA3 and 19 hours on Qwen. These results indicate that EPiC generalizes well across base models.

Table A3: Performance comparison between full-dataset training and EPiC on the OpenR1Math dataset across different backbone models, with similar format as Table 2.

| Methods | Math500 | | AIME24 | | AIME25 | | GPQA Diamond | | Time |
|---------|---------|-------|--------|--------|--------|--------|-----|--------|--------|
| | Acc | #Toks | Acc | #Toks | Acc | #Toks | Acc | #Toks | (Hours) |
| QWEN2.5-7B-INSTRUCT | | | | | | | | | |
| w/o SFT | 76.4 | 583.0 | 10.0 | 1061.8 | 3.3 | 921.7 | 30.8 | 577.8 | - |
| Full dataset | 84.4 | 3499.4 | 26.7 | 7590.1 | 20.0 | 7649.2 | 35.9 | 3798.7 | 52.1 |
| EPiC | 84.2 | 3378.4 | 26.7 | 7839.1 | 23.3 | 7476.6 | 35.9 | 3760.1 | 33.2 |
| LLaMA3.1-8B-INSTRUCT | | | | | | | | | |
| w/o SFT | 47.4 | 1311.7 | 3.3 | 3180.6 | 0.0 | 2779.6 | 29.8 | 1060.4 | - |
| Full dataset | 78.2 | 9000.0 | 13.3 | 9000.0 | 23.3 | 9000.0 | 27.2 | 4000.0 | 56.6 |
| EPiC | 75.0 | 9000.0 | 16.7 | 9000.0 | 10.0 | 9000.0 | 29.8 | 4000.0 | 37.5 |

## F    VISUALIZATIONS OF MODEL RESPONSES AFTER TRAINING WITH EPiC

To qualitatively evaluate the effectiveness of EPiC, we present representative examples of model-generated responses from the AIME24 benchmark in Table A4. The example is generated by a model fine-tuned using the condensed dataset produced by EPiC.In the visualization, the input question ($\mathbf{x}$), the reasoning trace ($\mathbf{r}$), and the final answer ($\mathbf{y}$) are shown. We highlight reflection cues and structural tokens (*e.g.*, `Wait`, `Therefore`, `<think>`, `</think>`) in **red** to indicate the model's internal reasoning structure. The correct final answer is marked in green. The visualization demonstrates that despite being trained on pruned traces that remove the middle portion of reasoning steps, the model retains coherent reasoning abilities and produces accurate answers, without hurting its capacity for reflection or disrupting its thinking structure. This supports that EPiC enables reasoning enhancement without compromising the model's reflective capacity or overall reasoning ability.

## G    LIMITATIONS

While EPiC presents a practical approach to reducing training costs through thought-level condensation, several limitations remain. First, it relies on a heuristics-based segmentation of CoT traces into

Table A4: Visualization of a model-generated response from the AIME24 benchmark, produced by a model fine-tuned using EPiC. **x** denotes the input question, **r** the generated reasoning trace, and **y** the final answer. **Red** highlights reflection cues (*e.g.*, Wait, Therefore) and structural tokens (¡think¿, ¡/think¿) marking the reasoning span. The correct final answer is shown in green.

| | |
|---|---|
| **x** | Let $A$, $B$, $C$, and $D$ be point on the hyperbola $\frac{x^2}{20} - \frac{y^2}{24} = 1$ such that $ABCD$ is a rhombus whose diagonals intersect at the origin. Find the greatest real number that is less than $BD^2$ for all such rhombi. |
| **r** | **\<think\>**Okay, so I have this problem about a rhombus ABCD on the hyperbola $x^2/20 - y^2/24 = 1$. The diagonals intersect at the origin, and I need to find the greatest real number less than BD² for all such rhombi. Hmm, let me start by recalling some properties of hyperbolas and rhombuses.\n\n First, ... **But wait**, the problem says the diagonals intersect at the origin. Since the diagonals of a rhombus bisect each other, that means the origin is the midpoint of both diagonals. ... **Wait**, BD is from B to D. Coordinates of B: ... **Therefore** , the greatest real number less than $BD^2$ for all such rhombi is 480. So, the final answer is \boxed{480}.\n\n**Final Answer**\n\boxed{480}**\</think\>** |
| **y** | ..., Thus, the greatest real number less than \($BD^2$\) for all such rhombi is \(\boxed{480}\) |

head, middle, and tail, which may not align with the true semantic structure of reasoning. Second, the condensation ratio is globally defined and does not adapt to the difficulty of individual examples, more challenging problems may benefit from retaining additional reasoning steps. Third, EPiC is implemented via supervised fine-tuning and does not explore reinforcement learning (RL)-based training, which could enable more dynamic, reward-driven condensation. Finally, our evaluation is limited to structured mathematical reasoning; extending EPiC to domains such as open-ended QA or legal reasoning requires further validation.

## H BROADER IMPACT

This work aims to improve the efficiency of reasoning supervision in large language models by condensing CoT-type reasoning traces. By reducing the length of *training* trajectories, our approach can lower computational costs and carbon footprint, making reasoning-enhanced model training more accessible to researchers and practitioners with limited resources. However, efficiency gains through condensation may come at the expense of preserving subtle but important reasoning patterns, potentially affecting model robustness, interpretability, or fairness. These potential trade-offs highlight the need for further investigation into how condensation impacts downstream performance across diverse tasks, domains, and user groups. We encourage future work to assess these dimensions and to develop techniques that balance efficiency with reliability and inclusivity.

## I THE USE OF LARGE LANGUAGE MODELS (LLMS)

We used LLMs solely for linguistic refinement, including grammar polishing and clarity improvements, and not for the creation of any ideas, analyses, or experimental results in this paper.

Table A1: Visualization of a training example with EPiC. **x** is the input question, **r** is the full CoT reasoning trace, and **y** is the final answer. The red-highlighted reasoning steps in **r** correspond to the middle portion that is pruned, while the head and tail segments are retained for training.

| | |
|---|---|
| **x** | A ship traveling along a river has covered $24 km$ upstream and $28 km$ downstream, ... Determine the speed of the ship in still water and the speed of the river. |
| **r** | **\<think\>**\nOkay, so I need to find the speed of the ship in still water and the speed of the river. Let me start by recalling that when a ship is moving upstream, its effective speed is the speed of the ship minus the speed of the river. Conversely, when moving downstream, its effective speed is the ship's speed plus the river's speed. \n\nLet me denote the speed of the ship in still water as \\( v \\) (in km/h) and the speed of the river as \\( r \\) (also in km/h). Then, the upstream speed would be \\( v - r \\), and the downstream speed would be \\( v + r \\).\n\nThe problem mentions three different journeys:\n\n1. 24 km upstream and 28 km downstream, taking half an hour less than the second journey.\n2. ... Let me first work on the first equation:\n\n\\( \frac{24}{v - r} + \frac{28}{v + r} = \frac{30}{v - r} + \frac{21}{v + r} - 0.5 \\)\n\nLet me subtract \\( \frac{24}{v - r} + \frac{28}{v + r} \\) from both sides, but maybe it's better to bring all terms to the left side:\n\\( \frac{24}{v - r} + \frac{28}{v + r} - \frac{30}{v - r} - \frac{21}{v + r} + 0.5 = 0 \\)\n\nSimplify the terms:\nFor the upstream terms: \\( 24 - 30 = -6 \\), so \\( \frac{-6}{v - r} \\)\nFor the downstream terms: \\( 28 - 21 = 7 \\), so \\( \frac{7}{v + r} \\)\nSo, the equation becomes:\n\n\\( \frac{-6}{v - r} + \frac{7}{v + r} + 0.5 = 0 \\)\n\nSimilarly, for the second equation:\n\n\\( \frac{24}{v - r} + \frac{28}{v + r} - \frac{15}{v - r} - \frac{42}{v + r} - 0.5 = 0 \\)\n\nSimplify:\n\nUpstream terms: \\( 24 - 15 = 9 \\), so \\( \frac{9}{v - r} \\)\n\nDownstream terms: \\( 28 - 42 = -14 \\), so \\( \frac{-14}{v + r} \\)\n\nThus:\n\n\\( \frac{9}{v - r} - \frac{14}{v + r} - 0.5 = 0 \\)\n\nNow, we have two equations:\n\n1. \\( -\frac{6}{v - r} + \frac{7}{v + r} + 0.5 = 0 \\)\n2. \\( \frac{9}{v - r} - \frac{14}{v + r} - 0.5 = 0 \\) ... ... \\( 2v = 20 \\) Ž1d2 \\( v = 10 \\)\n\nSubstituting back into \\( v + r = 14 \\):\n\n\\( 10 + r = 14 \\) Ž1d2 \\( r = 4 \\)\n\nSo, the speed of the ship in still water is 10 km/h, and the speed of the river is 4 km/h.\n\nLet me verify these results by plugging them back into the original time equations.\nFirst journey: 24/(10-4) + 28/(10+4) = 24/6 + 28/14 = 4 + 2 = 6 hours.\nSecond journey: 30/(10-4) + 21/(10+4) = 30/6 + 21/14 = 5 + 1.5 = 6.5 hours. So, 6.5 - 0.5 = 6 hours, which matches the first journey. Good.\n\nThird journey: 15/(10-4) + 42/(10+4) = 15/6 + 42/14 = 2.5 + 3 = 5.5 hours. Then, 5.5 + 0.5 = 6 hours, which also matches the first journey. Perfect.\n\nTherefore, the solution is correct.\n\n**Final Answer**\nThe speed of the ship in still water is \boxed{10} km/h and the speed of the river is \boxed{4} km/h.\n **\</think\>** |
| **y** | **\*\*\*Fianl Answer\*\*\***, ..., The speed of the ship in still water is \\(\boxed{10}\\) km/h and the speed of the river is \\(\boxed{4}\\) km/h. |

