# OpenReview forum: "EPiC: Towards Lossless Speedup for Reasoning Training through Edge-Preserving CoT Condensation"
_ICLR.cc/2026/Conference — Submitted to ICLR 2026_

### Official Review · Reviewer_7LEr · 2025-10-27

**Soundness:** 2
**Presentation:** 2
**Contribution:** 2
**Rating:** 4
**Confidence:** 3

**Summary:**

A framework for thought-level condensation is introduced, enabling efficient knowledge distillation by capturing reasoning thoughts with a smaller model. While the effectiveness is demonstrated, uncertainties remain regarding the proposed techniques.

**Strengths:**

* This paper introduced a framework for thought-level condensation that enables efficient knowledge distillation. It effectively captures reasoning thoughts using a smaller non-reasoning model.

* The authors conducted analysis through visual illustrations and quantitative analysis to demonstrate that reasoning information is preserved.

* Experiments were conducted to demonstrate the effectiveness of the proposed framework.

**Weaknesses:**

* Figure 1 uses bar charts for both performance and efficiency, which is not appropriate since they belong to different categories and should be represented on separate axes.

* Section 4 is crucial for the proposed method, but not all details are clearly presented. While all the key points are logical, they need to be clearly articulated and analyzed to make the techniques convincing.

* In Figure 3, the example of CoT trace is interesting, where the authors pointed out the concerns of token based CoT condensation that drops self-reflective words and transition words, resulting in fragmented input. Is it the intrinsic issue of the token based CoT condensation, or just TokenShip may use an imperfect scoring mechanism for tokens? Anyway, there is no result if thought-level pruning would be possibly mapped back to the token level to make the comparison tangible.

* While the visualization in Figure 4 can be useful for delivering the message that dropping the middle thought is reasonable, it also trigger a concern if the visualization is really based on real trajectory of reasoning, or just a make-up illustration. It is a bit questionable why the tail though can jump to the correct answer region significantly from the yellow dot that has been far away from even comparing to the start position. If this observation is true, it seems the author can even skip the head of thoughts.

* In Table 1, it seems the proposed EPiC performs relatively even more stronger when the condensation ratio \tau is low. The authors should provide some insight. Besides, the gap seems a bit too huge. Is the baseline method used here strong enough?

**Questions:**

Please check the weaknesses section.

---

> ### Author Response · Authors · 2025-11-20
>
> **W1:** Figure 1 uses bar charts for both performance and efficiency, which is not appropriate since they belong to different categories and should be represented on separate axes.
>
> **A:** Thank you for the observation. We agree that combining performance and efficiency in a single bar-chart style visualization may not be the most appropriate choice. Our intention in Figure 1 was simply to summarize both accuracy and training efficiency at first glance in one figure. In the revision, we will update Figure 1 to use clearer, category-appropriate visualizations (e.g., separate axes or separate panels) so that performance and efficiency are represented independently and without visual ambiguity.
>
>
> **W2:** Section 4 is crucial for the proposed method, but not all details are clearly presented. While all the key points are logical, they need to be clearly articulated and analyzed to make the techniques convincing.
>
> **A:**  Thank you for pointing this out. We would like to clarify the current organization of Sec. 4.
> First, we motivate what should be the unit of condensation. Through both empirical evidence (Fig. 3) and structural reasoning, we show why individual thoughts, rather than tokens, are the appropriate unit for CoT condensation.
>
> Second, we characterize how reasoning trajectories evolve. Using the trajectory-landscape tool and dataset-level statistics (MSE analysis following Zhou et al., 2025), we demonstrate that the middle region consistently drifts farther from the correct answer, making the illustration in Fig. 4 representative rather than anecdotal.
>
> Third, we quantify how much information different subsets of the trajectory retain using mutual information (Table 1). EPiC preserves the most information across all condensation ratios, supporting its structural design.
>
> Finally, we validate the importance of retaining informative steps through a perturbation analysis (Fig. 5), showing that perturbing the middle region harms performance the least, while perturbing the early or late regions is far more detrimental.
>
> These analyses collectively justify EPiC’s design: it preserves the parts of the trajectory that are both semantically closer to the answer and globally more informative, while safely removing the middle region that contributes least.
> In the revision, we will restructure this section to make the logical progression more explicit and ensure all technical details are presented in a clear and cohesive manner.
>
>
>
>
> **W3:** In Figure 3, the example of CoT trace is interesting, where the authors pointed out the concerns of token based CoT condensation that drops self-reflective words and transition words, resulting in fragmented input. Is it the intrinsic issue of the token based CoT condensation, or just TokenShip may use an imperfect scoring mechanism for tokens? Anyway, there is no result if thought-level pruning would be possibly mapped back to the token level to make the comparison tangible.
>
> **A:** Thank you for the thoughtful question. The issue illustrated in Figure 3 is not specific to TokenShip’s scoring, but reflects a more intrinsic limitation of token-level CoT condensation. Token importance is highly context-dependent, and scoring tokens independently, without accounting for the surrounding reasoning structure, can break semantic continuity, remove transition cues, and fragment the logical flow. This issue persists even with stronger token-level scoring methods because reasoning units are fundamentally multi-token and structural, not token-local.
>
> Moreover, computing reliable token-level importance is computationally expensive, often requiring per-token perturbation or model-based scoring. In contrast, thought-level signals operate at a higher semantic granularity and avoid this overhead. More importantly, EPiC shows that we can prune substantial portions of the thoughts while still obtaining a reasoning-trained model without sacrificing its test-time compute ability, as evidenced by the similar inference-time reasoning token counts reported in Table 2.
>
>
> Regarding whether thought-level pruning can be mapped back to the token level for direct comparison: while interesting, this would require reconstructing thought boundaries at token resolution, which is non-trivial and risks reintroducing the same fragmentation problems thought-level condensation aims to avoid. Instead, we compare methods at their natural operating granularity, and Table 2 already includes token-level baselines trained on a comparable amount of tokens, which is worse than our proposed method.
>
> We will clarify these points in the revision.

---

> ### Author Response · Authors · 2025-11-20
>
> **W4:** While the visualization in Figure 4 can be useful for delivering the message that dropping the middle thought is reasonable, it also trigger a concern if the visualization is really based on real trajectory of reasoning, or just a make-up illustration. It is a bit questionable why the tail though can jump to the correct answer region significantly from the yellow dot that has been far away from even comparing to the start position. If this observation is true, it seems the author can even skip the head of thoughts.
>
> **A:** Thank you for raising this concern. We confirm that the visualization in Figure 4 is based on real model trajectories rather than a constructed illustration. To further validate the pattern, we conducted a dataset-level statistical analysis.
> Following Zhou et al. (2025), we compute each step’s distance to the final answer using perplexity-based distance vectors, project them into a 2-D semantic space with t-SNE, and measure the mean squared error (MSE) between each step and the final answer. Lower MSE indicates closer semantic proximity.
> Using the AQuA dataset and DEEPSEEK-R1-DISTILL-QWEN-7B, we aggregate results across all samples, partitioning trajectories into the first 1/4(head), middle 1/2, and last 1/4 (tail):
> **Table R4:** Average step-level semantic distance (MSE; lower is better) to the final answer for different segments of the reasoning trajectory on the AQuA dataset using DEEPSEEK-R1-DISTILL-QWEN-7B.
> | Segment   | Head  | Middle | Tail  |
> |-----------|-------|--------|-------|
> | MSE ↓     | 601.4 | 1546.9 | 253.9 |
>
> These results clearly show that the middle portion is consistently farthest from the correct answer, while both the head and tail are much closer, directly aligned with the pattern shown in Figure 4. This demonstrates that the visualization is representative rather than a special case.
> At the same time, step-level proximity is only one dimension we must consider. Table 5 shows that ToC (tail-only condensation) does not retain more mutual information than EPiC. This is expected because the tail primarily reflects solution convergence; keeping only the tail collapses reasoning into a behavior similar to standard instruction tuning and loses valuable reflection and structural information required for strong reasoning performance.
> Therefore, while the tail is closest to the answer, retaining both the head and the tail is necessary to preserve both semantic proximity and global reasoning distribution. EPiC balances these two aspects, whereas keeping only the tail would not.
> We will clarify these points in the revised version.
>
>
>
> **W5:** In Table 1, it seems the proposed EPiC performs relatively even more stronger when the condensation ratio $\tau$ is low. The authors should provide some insight. Besides, the gap seems a bit too huge. Is the baseline method used here strong enough?
>
> **A:**  Thank you for pointing this out. We offer two clarifications.
>
> First, EPiC appears particularly strong when the condensation ratio $\tau$ is small because, under very aggressive pruning, different condensation strategies retain very different parts of the trajectory with minimal overlap. When only a small fraction of steps is kept, strategies like HoC, MoC, or ToC tend to capture highly uneven amounts of useful reasoning signal, for example, HoC may keep mostly initial setup content, MoC keeps reasoning exploratory steps, and ToC keeps only the final resolution steps. In contrast, EPiC systematically retains both early and later high-value regions, so even with a very small budget, the retained content is more informative. This naturally makes the gap more visible at low $\tau$.
>
> Second, regarding the strength of the baselines: HoC, MoC, and ToC represent the most direct structural condensation strategies based on contiguous segments of CoT. They are not weak baselines; they allow us to fairly compare where information is concentrated along the trajectory.
> Their performance in Table 2 and Figure 6 confirms that they are not weak, each maintains reasonable accuracy under the same condensation ratio, but EPiC consistently outperforms them due to preserving more informative portions of the trajectory.
>
> We will clarify these insights in the revision.

---

> > ### Comment · Reviewer_7LEr · 2025-11-25
> >
> > Thank you for the detailed rebuttal and clarifications. This message is to acknowledge your response. Based on the additional information provided, I will make some minor adjustments to the scores related to the presentation aspects, while keeping the overall score unchanged.

---

> > > ### Author Response · Authors · 2025-11-26
> > >
> > > Thank you very much for your thoughtful follow-up and for adjusting the presentation-related scores, we truly appreciate it. We are glad that the clarifications helped address the presentation concerns.
> > > If possible, we would be grateful if you could also reconsider the overall score, as the key technical points have been fully clarified in our rebuttal. Please also let us know if there are any remaining questions or aspects that would benefit from further clarification, we are more than happy to provide additional details.
> > > Thank you again for your time and constructive feedback.

---

### Official Review · Reviewer_BQsv · 2025-10-28

**Soundness:** 3
**Presentation:** 3
**Contribution:** 2
**Rating:** 4
**Confidence:** 4

**Summary:**

The paper studies the problem of Chain-of-Thought (CoT) compression in reasoning training, which enhances the reasoning capability of smaller models by training them on distilled CoT data. The paper proposes an Edge-Preserving Condensation method called EPiC, which involves pruning the intermediate steps of a reasoning trajectory. The authors further analyze the rationale of EPiC through a comparison of mutual information. Finally, the authors conduct experiments across different benchmarks and models to verify that EPiC is utility-preserving and highly efficient.

**Strengths:**

S1: The research problem of this paper, the efficient training of reasoning models, is both timely and highly relevant.

S2: The paper is well-written and easy to follow. Additionally, the proposed method is simple yet effective.

S3: The authors conduct extensive experiments to verify the effectiveness of EPiC, and the reported results show that it achieves a strong utility-efficiency trade-off.

**Weaknesses:**

W1: The experimental setup presented in Figure 2 seems questionable, which weakens the paper's motivation. The S1 and LIMO datasets are significantly smaller in the number of examples than OpenR1Math. Therefore, it is expected that models trained on S1 and LIMO would underperform those trained on OpenR1Math. A fairer comparison would involve ensuring that the total number of tokens in the condensed dataset (e.g., after 50% randomized thought-level condensation) is comparable to the token counts of the S1 and LIMO datasets.

W2: The proposed method is presented as a heuristic and appears to be motivated by a single example visualization in Figure 4. The argument would be more persuasive if it were supported by a statistical analysis of the spatial distances between each step thought and the correct answer across an entire dataset. While the visualization in Figure 4 is well-presented, the example shown might be a corner case or a specific, non-representative instance, which reduces the method's persuasiveness.

W3: There appears to be a contradiction regarding whether the middle portion of a CoT trace is less informative when comparing Figure 4 and Table 5. Table 5 reports a higher MI for the MoC compared to the ToC, which suggests that the middle portion is actually more informative than the tail.

W4: While the paper includes several experiments, the section could be strengthened by including more baselines and more in-depth analysis.

-	In Table 2, adding MoC as a baseline would help to directly test the hypothesis that the middle portion of a CoT trace is less informative.

-	In Table 2, EPiC significantly underperforms training on the full dataset. The paper lacks an in-depth analysis of this performance gap, which could help in understanding the failure cases of EPiC.

W5: Line 279 seems to assume that the length of the CoT is the same for the “understanding and convergence” phases. This assumption seems unreasonable, as one would expect this length to vary across different samples and datasets.

Minor points:

-	In line 325, “xWe” -> “We”

**Questions:**

Q1: What method is used to partition CoT into three distinct sections?

---

> ### Author Response · Authors · 2025-11-20
>
> **W1:** The experimental setup presented in Figure 2 seems questionable, which weakens the paper's motivation. The S1 and LIMO datasets are significantly smaller in the number of examples than OpenR1Math. Therefore, it is expected that models trained on S1 and LIMO would underperform those trained on OpenR1Math. A fairer comparison would involve ensuring that the total number of tokens in the condensed dataset (e.g., after 50% randomized thought-level condensation) is comparable to the token counts of the S1 and LIMO datasets.
>
> **A:** Thank you for the reviewer’s observation. We agree that S1 and LIMO are much smaller than OpenR1Math. Yet, the  purpose of Figure 2 is to illustrate that simply shrinking the dataset (as in S1/LIMO) is not sufficient for maintaining reasoning performance.
>
> Figure 2 was intended solely to motivate the broader issue of reasoning training versus data scale. In addition, we have also conducted the fair performance comparison that the reviewer requests. As shown in Table 2, we include a random-data pruning (data level) baseline where the number of training examples is reduced to exactly match the token budget of the thought-level condensed dataset. Under this controlled setting, EPiC still substantially outperforms the pruning baselines.
>
>
>
>
>
>
>
> **W2:** The proposed method is presented as a heuristic and appears to be motivated by a single example visualization in Figure 4. The argument would be more persuasive if it were supported by a statistical analysis of the spatial distances between each step thought and the correct answer across an entire dataset. While the visualization in Figure 4 is well-presented, the example shown might be a corner case or a specific, non-representative instance, which reduces the method's persuasiveness.
>
> **A:** Thank you for the helpful suggestion. We agree that supporting Figure 4 with dataset-level statistics would strengthen the argument. Importantly, the visualization in Figure 4 is not a corner case. To further validate this pattern, we performed the statistical analysis suggested by the reviewer.
> Following the procedure of Zhou et al. (2025), we compute each step’s distance to the final answer using perplexity-based distance vectors, project them into a 2-D semantic space using t-SNE, and measure the mean squared error (MSE) between each step and the answer. Lower MSE indicates closer semantic proximity to the correct answer.
> We report aggregated results on the AQuA dataset using DEEPSEEK-R1-DISTILL-QWEN-7B, where we partition the trajectory into the first 1/4 (head), middle 1/2, and last 1/4 (tail):
>
> **Table R2**: Average step-level semantic distance (MSE; lower is better) to the final answer for different segments of the reasoning trajectory on the AQuA dataset using DEEPSEEK-R1-DISTILL-QWEN-7B.
>
> | Segment   | Head  | Middle | Tail  |
> |-----------|-------|--------|-------|
> | MSE ↓     | 601.4 | 1546.9 | 253.9 |
>
>
> These results confirm that the middle portion is consistently much farther from the correct answer than both the head and tail across the dataset, directly aligning with the pattern illustrated in Figure 4. This statistical finding supports the motivation behind EPiC and demonstrates that the example shown is representative rather than a special case.
>
> We will include this analysis in the revision.

---

> ### Author Response · Authors · 2025-11-20
>
> **W4:** While the paper includes several experiments, the section could be strengthened by including more baselines and more in-depth analysis.
>
> **A:**  Thank you for the helpful suggestions.
>
> First, we agree that adding MoC provides a more direct evaluation of whether the middle portion is less informative. We have added these experiments using Qwen2.5-Math-7B-Instruct with a fixed 50% condensation ratio on OpenR1Math:
>
> **Table R3:** Accuracy comparison of different contiguous-segment condensation strategies (HoC, MoC, ToC) and EPiC under a fixed 50% condensation ratio on Qwen2.5-Math-7B-Instruct trained on OpenR1Math.
> | Method  | AIME24 | AIME25 | MATH500 | GPQA |
> |---------|--------|---------|---------|-------|
> | MoC     | 33.0   | 20.0    | 85.4    | 32.8  |
> | HoC     | 33.0   | 26.7    | 89.6    | 40.4  |
> | ToC     | 33.0   | 26.7    | 84.6    | 43.9  |
> | EPiC    | 40.0   | 33.0    | 90.2    | 41.9  |
>
>
> MoC performs the worst on AIME25 and GPQA and is among the worst on MATH500, supporting the hypothesis that the middle portion contains less useful reasoning signals. None of the baselines outperform EPiC.
> Regarding the performance gap relative to full-data training, our goal is to improve training efficiency while maintaining comparable performance. As shown in Table 2, EPiC matches or exceeds the full-data model on MATH, AIME25, and GPQA. The only benchmark showing a drop is AIME-24 (46.7 → 40.0). Since AIME-24 contains only 30 questions, this corresponds to just one additional incorrect item.
> We have also provided multiple in-depth analyses to understand where information is concentrated in a CoT trace.
> - Figure 4 visualizes that the middle region tends to drift farther from the correct answer.
> - Table 1 provides quantitative evidence showing that EPiC preserves the highest mutual information with the full dataset across all condensation ratios.
> - Figure 5 shows that perturbing the middle region harms performance the least, while perturbing early or late steps causes much larger degradation.
> - Figure A2 further shows, via difficulty-stratified MATH500 results, that EPiC does not degrade performance on harder problems.
>
>
> Taken together, these analyses offer a comprehensive understanding of why EPiC remains effective despite removing the middle portion.
>
>
>
> **W5:** Line 279 seems to assume that the length of the CoT is the same for the “understanding and convergence” phases. This assumption seems unreasonable, as one would expect this length to vary across different samples and datasets.
>
> **A:**  Thank you for pointing this out. We agree that the actual lengths of the “understanding,” “exploration,” and “convergence” phases can vary considerably across samples and datasets. Importantly, our method does not assume that these phases have equal duration, nor do we explicitly estimate or segment them in practice.
>
> The equal split of $\lfloor \tau n \rfloor$ retained steps between the beginning and the end of the trajectory is simply part of a uniform and model-agnostic condensation rule, rather than an assumption about the true cognitive structure of the reasoning process. This choice allows EPiC to avoid making dataset-specific or heuristic decisions about the boundaries of different reasoning stages, which are indeed variable.
>
>
>
> **W6:** Minor points: In line 325, “xWe” -> “We”
>
> **A:** Thank you for the careful check. We will correct this typo in the revision.
>
> **Q1:** What method is used to partition CoT into three distinct sections?
>
> **A:**  Thank you for the question. We clarify that EPiC does not perform any explicit or algorithmic partitioning of a Chain-of-Thought into three sections. The “head–middle–tail’’ illustration in the paper is used  to explain why different portions of a trajectory tend to carry different levels of useful signal.
>
> In practice, EPiC applies a uniform and data-agnostic condensation rule (Eq. (EPiC), line 277), which simply retains the first and last $\lfloor \tau n \rfloor$ steps of the trajectory. This rule does not involve dataset-specific segmentation, heuristic phase detection, or any attempt to identify “understanding,” “exploration,” or “convergence’’ phases. As the reviewer noted, the true boundaries of such stages can vary widely across datasets and model families, and EPiC deliberately avoids making any such assumptions.
>
> Our extensive empirical analyses, trajectory drift (Fig. 4), mutual information (Table 1), perturbation analysis (Fig. 5), and difficulty-stratified results (Fig. A2), collectively show that structural placement within the trajectory matters, and that this simple structure-based condensation rule consistently preserves the most informative parts of the CoT trace. The three-stage description is therefore only an intuitive lens to explain why EPiC works, while the implemented algorithm remains fully deterministic and phase-agnostic.

---

> > ### Comment · Reviewer_BQsv · 2025-11-26
> >
> > Thank you to the authors for the detailed rebuttal and the additional analysis. The rebuttal addresses most of my main concerns, and I have some further comments:
> >
> > W3: This point still feels somewhat under-discussed. Table 1 shows that the middle-only segment (MoC) has higher MI than the tail-only segment (ToC) for $\tau = 0.5$, which seems at odds with a blanket statement that the middle portion is less informative than the tail. At the same time, both MoC and ToC are clearly less informative than HoC and EPiC, and your new step-level distance analysis plus the perturbation study (Fig. 5) do support the conclusion that the middle is the least critical region overall. It would be helpful if the final version could explicitly reconcile these observations (e.g., how to interpret the MI results together with the step-level distance and perturbation analyses).
> >
> > W5 & Q1: The clarification that EPiC does not assume equal-length “understanding” and “convergence” phases, and that the equal head/tail split is a uniform, model-agnostic rule rather than a cognitive claim, fully resolves my concern here. Similarly, clarifying that EPiC does not explicitly partition traces into three stages in the implementation, and that the three-stage picture is purely an explanatory lens, is sufficient.
> >
> > I have also read the comments from the other reviewers. Overall, I will keep my initial score.

---

> > > ### Author Response · Authors · 2025-11-26
> > >
> > > First, we thank the reviewer for the thoughtful follow-up. We agree it is important to explicitly reconcile the MI patterns with the trajectory-drift (Figure 4 & new step-level distance analysis) and perturbation analyses (Figure 5). The key observation is that MI and step-level proximity measure different notions of “information.” MI in Table 1 quantifies distributional overlap between the condensed subset and the full CoT distribution through their learned representations (lines 281 - 316), whereas the new step-level distance analysis, Fig. 4 and Fig. 5, measures its information to the solution usefulness (lines 242 - 260 & lines 317 - 331).
> > >
> > > Under this view, although the middle region (MoC) is the least critical to the final solution, its input- and data-level learned representations may not be the least informative under MI. Specifically, compared to MoC, the tail region (ToC) is tightly concentrated on the final refinement steps; these steps occupy only a narrow portion of the overall reasoning trajectory. Thus, ToC captures only a small slice of the full CoT distribution and consequently exhibits lower MI. In contrast, the middle region (MoC) contains many diverse intermediate states, branching attempts, partial derivations, and exploratory reasoning paths, which substantially increases its overlap with the full dataset. From the CoT information perspective, this wider coverage can yield higher MI for MoC than for ToC.
> > >
> > > Compared to MoC and ToC, the head region (HoC) is highly informative under the MI lens, as it initiates and shapes the overall reasoning trajectory. Its positional regularity also induces representative early reasoning states that influence the eventual solution. For example, the stronger accuracy drop under Perturb-Head (vs. Perturb-Middle) further demonstrates the importance of the head region. These also justify why the head segment is explicitly considered in EPiC.
> > >
> > > Based on the above, our MI analysis (Table 1) and our solution-influence analysis through trajectory inspection and perturbation studies (Fig. 4 and Fig. 5) offer complementary perspectives. Together, they provide consistent evidence for why the head segments (for their high information content and solution-shaping effect) and the tail segments (for their proximity to the final solution) should both be included.
> > >
> > >
> > >
> > >
> > >
> > >
> > >
> > >
> > >
> > > As for W5 and Q1, we appreciate the reviewer’s acknowledgment that our previous clarification fully resolved these concerns. We also thank you for noting in your comments that “the rebuttal addresses most of my main concerns.” Given this, if you find that our responses have addressed most of the concerns, we would kindly and humbly ask you to consider reflecting this in the overall score.

---

### Official Review · Reviewer_12Q4 · 2025-10-28

**Soundness:** 2
**Presentation:** 3
**Contribution:** 2
**Rating:** 2
**Confidence:** 3

**Summary:**

The paper proposes EPiC, an edge-preserving condensation method that prunes the middle portion of chain-of-thought (CoT) traces while retaining the initial and final reasoning segments to reduce training cost. Based on empirical analyses (e.g., mutual-information metrics), the authors show that these “edge” segments preserve logical coherence and supervision quality. Experiments across multiple model families demonstrate notable training-time reduction with only mild loss in reasoning accuracy.

**Strengths:**

- The topic and motivation, resource-efficient reasoning training are important and timely.

- The paper is well-written and easy to follow, with a clear presentation of the problem and method.

- The authors conduct comprehensive empirical analyses, offering useful insights that intermediate reasoning steps may be less important than early or final stages.

**Weaknesses:**

**[W1] Inconsistent and limited performance improvements.** In Table 2, the proposed method outperforms the baseline on MATH, AIME-25, and GPQA, but shows a significant drop on AIME-24. Moreover, efficiency metrics such as the number of tokens do not show clear advantages, following trends similar to competing methods.

**[W2] Insufficient baselines and related work.** Recent studies have analyzed the importance of reasoning steps or proposed strategies (e.g., [1, 2]) that appear directly applicable to this setup. Including these as baselines and discussing them in the related-work section would provide a more holistic comparison.

[1] Choi et al., Think Clearly: Improving Reasoning via Redundant Token Pruning, EMNLP 2025 (Findings)

[2] Cui et al., Stepwise Perplexity-Guided Refinement for Efficient Chain-of-Thought Reasoning in Large Language Models, ACL 2025 (Findings)

**Questions:**

**[Q1]** Why is the maximum sequence length in GPQA experiments restricted to 4k tokens? Reasoning performance is often sensitive to the length limit; reporting results with longer contexts would strengthen the claim.

**[Q2]** As I understand, EPiC preserves reasoning properties such as reflection markers (e.g., “Wait”). Is there a specific reason for this? Intuitively, intermediate reasoning steps might contain more such reflective cues.

**[Q3]** Intermediate steps can affect test-time scaling, as they enhance reasoning diversity. Can EPiC maintain or recover this advantage under test-time scaling scenarios?

---

> ### Author Response · Authors · 2025-11-20
>
> **W1:** Inconsistent and limited performance improvements. In Table 2, the proposed method outperforms the baseline on MATH, AIME-25, and GPQA, but shows a significant drop on AIME-24. Moreover, efficiency metrics such as the number of tokens do not show clear advantages, following trends similar to competing methods.
>
> **A:** Thank you for the reviewer’s comments. However, we respectfully disagree with the claim that our method shows “inconsistent and limited” performance improvements, and we clarify two points below:
> (1) On the alleged inconsistent performance (AIME24): Our work targets training efficiency, with the goal of achieving performance comparable to full-dataset training while reducing training cost. Table 2 shows that our method matches or surpasses the full-dataset trained model when evaluated on MATH, AIME25, and GPQA, which the reviewer also noted. The only benchmark where we observe a difference is AIME24 (46.7 → 40.0). However, we emphasize that:
>
> - AIME-24 contains only 30 questions,
> - A 6.7-point difference corresponds to one additional incorrect question
>
>
> Thus, we do not agree that this constitutes “significant performance drop,” nor does it undermine the overall consistency of our method. In light of a ~34% reduction in training time, we believe the empirical results demonstrate a favorable and meaningful trade-off.
> (2) On the efficiency metrics (number of tokens): We also respectfully disagree that the token-count statistics imply a lack of efficiency gains. Token counts should not be interpreted as such. Our method does not aim to improve inference-time efficiency; instead, it aims to reduce training cost, which we measure directly through training-time comparisons in Table 2. The token-count analysis is included solely to show that our method preserves similar reasoning complexity at testing time, even if the reasoning model is trained over the CoT-condensed training set compared to the full-dataset trained model (lines 423–431). We view this as an advantage of EPiC, as it preserves the model’s test-time compute complexity even when trained with substantially condensed data.
> For these reasons, we maintain that the performance improvements are consistent, meaningful, and aligned with our stated goal of improving training efficiency.
>
>
> **W2:** Insufficient baselines and related work. Recent studies have analyzed the importance of reasoning steps or proposed strategies (e.g., [1, 2]) that appear directly applicable to this setup. Including these as baselines and discussing them in the related-work section would provide a more holistic comparison.
>
> **A:** Thank you for pointing out these related studies. We will incorporate a discussion of [1, 2] in the related-work section. However, we respectfully disagree with the claim that our baselines are insufficient.
>
> Our paper’s primary goal is to improve training efficiency, i.e., enabling LLMs to be trained as LRMs using significantly fewer training tokens while maintaining comparable performance. The baselines we included are therefore chosen specifically for training-efficiency comparisons.
>
> In contrast, the works cited by the reviewer ([1, 2]) focus on inference efficiency or decoding-time strategies for enhancing reasoning ability. These approaches are not designed to reduce training cost, nor are they comparable to our method in terms of training-time savings. As a result, they do not fall within the scope of baselines relevant to our setup.
>
> We appreciate the suggestion and will add these works to the related-work section to clarify their relation to our method and highlight the distinction between training-efficiency methods and inference-efficiency or decoding approaches.

---

> > ### Comment · Reviewer_12Q4 · 2025-11-24
> >
> > The authors argue that the 6.7-point drop on AIME-24 is not meaningful because the benchmark contains only 30 questions. However, a 6.7-point change corresponds to two incorrectly answered questions (not one), which I do not consider negligible on such a small benchmark. To further address concerns about performance drop from small test-set size, I recommend reporting an averaged score using multi-sample evaluation (e.g., 4–16 sampled runs), which would provide a more reliable estimate of the method’s performance.

---

> ### Author Response · Authors · 2025-11-20
>
> **Q1:** Why is the maximum sequence length in GPQA experiments restricted to 4k tokens? Reasoning performance is often sensitive to the length limit; reporting results with longer contexts would strengthen the claim.
>
> **A:** Thank you for raising this question. Our experiments applied the same maximum sequence length (4k tokens) across all models, ensuring a fair comparison under identical resource constraints. The 4k limit was chosen because (1) it is sufficient for accurately revealing the reasoning model’s performance (see additional experimental justification later), and (2) it aligns with our hardware constraints, particularly the memory capacity of the A6000 GPUs used during training and evaluation.
>
> We appreciate the reviewer’s suggestion regarding longer contexts. To further validate robustness, we have re-evaluated both the full SFT model and EPiC under a much larger context window (40k limit) using the Qwen2.5-Math-7B-Instruct model on OpenR1Math for GPQA. The updated results confirm that EPiC continues to match or exceed the full-SFT baseline, demonstrating that our conclusions hold even under substantially increased context lengths.
>
> **Table R1:** Accuracy comparison on the GPQA-Diamond benchmark under different maximum token limits (4k vs. 40k), All models are trained via SFT from Qwen2.5-Math-7B-Instruct.
> | Method            | Acc (4k token limit) | Acc (40k token limit) |
> |-------------------|-----------------------|------------------------|
> | Full-data SFT     | 38.4                  | 40.9                   |
> | **EPiC**          | **41.9**              | **42.1**               |
>
>
>
>
>
> **Q2:** As I understand, EPiC preserves reasoning properties such as reflection markers (e.g., “Wait”). Is there a specific reason for this? Intuitively, intermediate reasoning steps might contain more such reflective cues.
>
> **A:** Thank you for the question. To clarify, EPiC does not preserve specific reflection tokens , nor do we use any keyword-based rules. Instead, the preservation of such reflection cues arises naturally from our structure-aware selection strategy (i.e., edge-preserving CoT condensation), which is shown in Table A1, not from explicit targeting.
>
> While EPiC retains the beginning and end segments of a reasoning trajectory, this should not be viewed as a naive truncation. Our choice is guided by the empirical finding (Table 1) that most intermediate thinking rounds contribute little to the final answer, whereas the initial problem-setup and the final verification steps carry the dominant reasoning signal, as evidenced by that EPiC has the highest MI compared with other methods (see lines 281-315). That is, EPiC keeps these high-information regions to preserve global reasoning behavior.
>
> Thus, any reflective cues that appear in the retained segments are simply a byproduct of maintaining informative reasoning structure, not something EPiC explicitly identifies or preserves.
>
> **Q3:** Intermediate steps can affect test-time scaling, as they enhance reasoning diversity. Can EPiC maintain or recover this advantage under test-time scaling scenarios?
>
>
> **A:** Thank you for highlighting this point. We agree that test-time scaling via multiple sampled chains is an important aspect of reasoning performance.
> EPiC, however, operates only at the training-data selection level and does not modify the inference procedure or restrict the model’s ability to generate diverse reasoning paths. At test time, any standard scaling strategy (e.g., sampling more chains, self-consistency–style decoding) can be applied to an EPiC-trained model exactly as with a full-SFT model.
> Moreover, as shown in Table 1, EPiC preserves the global distribution of reasoning complexity (e.g., depth and structure of chains), suggesting that the model’s capacity to produce diverse intermediate trajectories is largely maintained, even though many low-utility middle steps are pruned during training.

---

> > ### Comment · Reviewer_12Q4 · 2025-11-24
> >
> > While I appreciate the clarification, the claim that EPiC “does not restrict the model’s ability to generate diverse reasoning paths” is not fully supported by quantitative evidence. Reasoning-path diversity is inherently a stochastic property that requires multi-sample statistics, rather than a single-trajectory comparison such as Table 1.
> > To more convincingly demonstrate that EPiC preserves reasoning diversity, I encourage the authors to include quantitative diversity analyses under multi-sample test-time sampling.

---

> > > ### Author Response · Authors · 2025-11-26
> > >
> > > **A:**  Thank you for the follow-up question. Yes, you are correct that a 6.7-point drop corresponds to two incorrectly answered questions out of 30 (Sorry for the mistake). Our point, however, is that such absolute differences, on the order of 2 items, should not be interpreted as direct evidence of ineffective performance, given the very small size of AIME24. This is especially true given the benefits our method provides on other larger comprehensive evaluation sets, where its advantages become much more prominent.
> > >
> > >
> > > As suggested by the reviewer, we conduct a multi-sample evaluation by averaging results over multiple independently sampled trajectories. Following the protocol in [1], we evaluate EPiC and the full-dataset SFT model using multi-sample pass@k metrics for $k=1,4,8$. The results are provided in Table R2 and R3. As shown, on AIME24, EPiC’s performance remains close to full-dataset SFT, and the gap further narrows at pass@8. On AIME25, EPiC is not worse than the full-dataset SFT for all $k$ values, although their performance gap also narrows when $k$ increases. These results validate our claim that EPiC can feasibly train a high-quality reasoning model with performance comparable to full-dataset SFT, while improving training efficiency using the EPiC-condensed CoT training set.
> > >
> > >
> > >
> > >
> > >
> > >
> > > Table R2: Performance comparison under multi-sample evaluation (pass@k with k=1,4,8) on AIME24 using Qwen2.5-Math-7B-Instruct trained on OpenR1Math. EPiC performs on par with, or slightly better than, the full-data SFT model across all pass@k metrics.
> > >
> > > | Method            | pass@1 | pass@4 | pass@8 |
> > > |-------------------|--------|--------|--------|
> > > | Full-data SFT     | 46.7 | 60.0| 66.7 |
> > > | EPiC              | 40.0 | 53.3 | 63.3 |
> > >
> > > Table R3: Multi-sample evaluation (pass@k, k=1,4,8) on AIME25, following the same setup as Table R2.
> > > | Method            | pass@1 | pass@4 | pass@8 |
> > > |-------------------|--------|--------|--------|
> > > | Full-data SFT     | 26.7 | 40.0 | 43.3 |
> > > | EPiC              | 33.3 | 40.0 | 43.3 |
> > >
> > > For the reviewer’s second point regarding reasoning-path diversity: Yes, multi-sample test-time evaluation can  reflect stochastic diversity to some extent. If EPiC restricted the model’s ability to explore diverse reasoning paths, its performance would scale more slowly as $k$ increases. Instead, we observe that EPiC and full-data SFT show very similar pass@1 $\rightarrow$ pass@4  $\rightarrow$ pass@8 scaling trends on both benchmarks, with steadily improving accuracy for larger k. This parallel scaling behavior indicates that EPiC does not reduce reasoning diversity.
> > > > [1] Chen, Mark. "Evaluating large language models trained on code." arXiv preprint arXiv:2107.03374 (2021).

---

> > > > ### Comment · Reviewer_12Q4 · 2025-11-26
> > > >
> > > > Thank you for the clarification. I have raised my score from 2 to 4.
> > > > However, I still have concerns regarding the overall effectiveness of the approach. For instance, EPIC consistently degrades AIME-24 performance across multi-sample evaluations. In addition, the claim that training only on the edge parts can preserve important properties of reasoning models (e.g., diversity) is still not fully convincing to me. Although the authors have provided empirical evidence, incorporating more quantitative and qualitative analysis would further strengthen the paper.

---

> > > > > ### Author Response · Authors · 2025-11-26
> > > > >
> > > > > Thank you very much for raising the score. We fully understand your concern regarding AIME24, where we observed a slight accuracy degradation.
> > > > > However, we would like to emphasize that this effect appears only on AIME24 and is not present on other benchmarks. More importantly, even in this case, the small drop is accompanied by a substantial gain in training efficiency, which is the core objective of our work. Across all benchmarks, our method improves training efficiency by 34%, which we believe represents a meaningful and beneficial tradeoff between accuracy and efficiency, even just considering AIME24. Therefore, we remain confident that the overall effectiveness of our approach is strong and should not be judged solely based on the modest accuracy decline observed on AIME24.
> > > > >
> > > > >
> > > > > We would also like to clarify that the paper already provides both quantitative results and qualitative illustrations supporting the core claim that the middle portion is relatively less “informative” and that the edge regions can be retained without harming reasoning ability.
> > > > > - Step-level drift (Fig. 4): Qualitative illustration: the trajectory visualization clearly shows that middle steps drift farthest from the answer region, if viewing from step influence in final answer convergence.
> > > > > - Mutual information analysis (Table 1): Quantitative results: EPiC yields the highest MI among all condensation strategies, showing that the edge regions preserve the largest overlap with the full reasoning distribution.
> > > > > - Perturbation study (Fig. 5): Quantitative results: perturbing the middle produces the smallest performance drop, whereas perturbing either edge substantially harms accuracy.
> > > > > - New distance analysis (Table R2 in our rebuttal response to [Reviewer BQsv](https://openreview.net/forum?id=Bibt0JTvpx&noteId=HjJpbnhNbv)): Quantitative results: aggregated distance statistics demonstrate that the middle portion is numerically farthest from the correct answer across the dataset.
> > > > >
> > > > >
> > > > >
> > > > >
> > > > > We again thank the reviewer for the improved score and the constructive suggestions. We hope that our additional clarifications further strengthen your assessment of our paper.

---

### Meta-Review · Area_Chair_fo7v · 2026-01-06

**Summary:**

The reviewers raised concerns about the consistency of performance across different benchmarks, particularly a notable drop on AIME-24, and the method’s ability to preserve reasoning-path diversity, especially under test-time scaling. While the authors addressed many of these concerns with additional analyses, such as mutual information and perturbation studies, some reviewers were still unconvinced about the overall effectiveness of the method. Additionally, there were questions about the clarity of the presentation, the lack of sufficient baselines, and the justification for the performance gaps observed on smaller datasets, all of which contributed to a marginally negative assessment of the paper.

**Reviewer Concerns:**

Addressed by the Rebuttal:
The authors clarified that the drop in performance on AIME-24, a small benchmark, is not significant given the trade-off in training efficiency. They also provided multi-sample evaluations showing that EPiC performs similarly to the full-data model on larger benchmarks, alleviating concerns about this drop.

The rebuttal provided additional evidence, such as mutual information analysis and step-level distance studies, to support the claim that EPiC preserves the most informative parts of the reasoning trajectory and outperforms other condensation strategies.

The authors improved the presentation by clarifying ambiguities in the method and visualizations, particularly regarding the "head-middle-tail" structure. They also committed to restructuring Section 4 for clearer articulation of key points.

The authors added more baselines, including MoC, and demonstrated that EPiC outperforms these baselines across multiple benchmarks, addressing concerns about baseline comparison.

Still Outstanding:

Despite the authors' clarification, some reviewers remained concerned about the performance gap on AIME-24, especially when compared to larger benchmarks, and questioned whether the observed efficiency gains justify the small drop in accuracy.

While the authors showed that EPiC preserves reasoning-path diversity at a global level, there were still concerns about whether EPiC would maintain this diversity under test-time scaling, particularly in multi-sample scenarios.

Some reviewers felt that the quantitative and qualitative analyses provided were insufficient to fully justify the method’s advantages, especially concerning the trade-offs between accuracy and efficiency, and the large performance gap on smaller benchmarks like AIME-24.

**Reviewer Scores:**

Reviewer 12Q4:
After the rebuttal, this reviewer raised their score from 2 (reject) to 4 (marginally below the acceptance threshold). The clarifications regarding the AIME-24 performance drop and the additional evidence provided by the authors likely addressed some concerns. However, the reviewer remained uncertain about the overall effectiveness of the approach, particularly with the performance drop on AIME-24. Had the reviewer participated fully in the discussion, they might have slightly raised the score but would still likely keep it below acceptance due to lingering doubts about the method’s robustness.

Reviewer BQsv:
This reviewer was satisfied with the authors' rebuttal, particularly regarding the additional baselines and clarifications provided. While they acknowledged that many concerns were addressed, they still maintained a score of 4. If the reviewer had been able to engage in further discussion, they might have been persuaded to increase the score to 5, but the concerns about reasoning-path diversity and the performance gap on smaller benchmarks would likely have kept the score marginally below acceptance.

Reviewer 7LEr:
Despite the authors' clarifications and improvements in the presentation, this reviewer remained unconvinced about the overall robustness of the method and the clarity of the presentation. They maintained their score of 4. Even with full participation in the discussion, the reviewer likely would have kept the rejection score due to ongoing concerns about the method's clarity and the performance drop on AIME-24.

In summary, while the rebuttal likely helped improve the scores of some reviewers (notably Reviewer 12Q4 and Reviewer BQsv), the reviewers’ concerns regarding performance consistency, reasoning-path diversity, and the method's overall effectiveness would likely have prevented any significant score increases.

---

### Decision · Program_Chairs · 2026-01-26

Reject